# Footprint-weighted tile approach for a spruce forest and a nearby patchy clearing using the ACASA model

Kathrin Gatzsche[1,2,*], Wolfgang Babel[1,3,**], Eva Falge[4,***], Rex David Pyles[5], Kyaw Tha Paw U[5], Armin Raabe[2], and Thomas Foken[1,3]

[1]University of Bayreuth, former Department of Micrometeorology, Bayreuth, Germany
[2]University of Leipzig, Leipzig Institute for Meteorology, Leipzig, Germany
[3]University of Bayreuth, Bayreuth Center of Ecology and Environmental Research (BayCEER), Bayreuth, Germany
[4]Max-Planck-Institute of Chemistry, Biogeochemistry Department, Mainz, Germany
[5]University of California, Department of Land, Air and Water Resources, Davis, California, USA
[*]now at: Leibniz Institute for Tropospheric Research (TROPOS), Department Modelling of Atmospheric Processes, Leipzig, Germany
[**]now at: University of Bayreuth, Micrometeorology Group, Bayreuth, Germany
[***]now at: German Meteorological Service, Agrometeorological Research Center, Braunschweig, Germany

*Correspondence to:* T. Foken (thomas.foken@uni-bayreuth.de)

**Abstract.** The ACASA (Advanced Canopy-Atmosphere-Soil Algorithm) model, with a higher order closure for tall vegetation, has already been successfully tested and validated for homogeneous spruce forests. The aim of this paper is testing the model using a footprint-weighted tile approach for a clearing with a heterogeneous structure of the underlying surface. The comparison with flux data shows a good agreement with a footprint aggregated tile approach of the model. However, the results of a comparison with a tile approach on the basis of the mean land use classification of the clearing is not significantly different. It is assumed that the footprint model is not accurate enough to separate small scale heterogeneities. All measured fluxes are corrected by forcing energy balance closure of the test data either by maintaining the measured Bowen ratio or by the attribution of the residual depending on the fractions of sensible and latent heat flux to the buoyancy flux. The comparison with the model – where the energy balance is closed – shows that the buoyancy correction for Bowen ratios $> 1.5$ better fits the measured data. For lower Bowen ratios, the correction probably lies between both methods, but the amount of available data was too small to make a conclusion. With an assumption of a similarity between water and carbon dioxide fluxes, no correction of the net ecosystem exchange is necessary for Bowen ratios $> 1.5$.

## 1 Introduction

The comparison of modeled and measured energy and matter fluxes in a heterogeneous landscape is still a challenge. The fluxes measured with the eddy-covariance technique are related to all surfaces on the up-wind side of the measurements, and the influence of each surface on the measured data is given by the footprint function. The comparison of these flux measurements with 1-dimensional models can be done over a homogeneous surface easily or over a heterogeneous surface if the model is parameterized for all surface characteristics and averaged according to the percentage of each surface in the footprint (tile

approach, Mölders, 2012). The footprint varies with changing wind velocity and atmospheric stratification for each half-hourly derived flux, consequently the tile approach must also be modified every half hour. This approach was proposed by Foken and Leclerc (2004) and successfully improved by Göckede et al. (2005) and Biermann et al. (2014) for two separated surfaces. The tile approach is based on the application of the eddy-covariance method with the correction of the storage term, which ignores

any advection. The more appropriate approach would use the generalized eddy-covariance method (Foken et al., 2012), which includes the advection of matter and fluxes. Unfortunately, matter advection could only be analyzed experimentally under very idealized conditions up to now. Experimental advection studies with an acceptable number of measurement points failed (Aubinet et al., 2010, 2012). Furthermore, the resolution of available 3-D models (Sogachev et al., 2002; Sogachev and Lloyd, 2004) is too large ($25-50$ m) for the small scale heterogeneities on the clearing ($5-10$ m). For our case, an experimental advection

experiment cannot be realized and only an LES process study could be applied for a single 1 hour case (Kanani-Sühring and Raasch, 2017). Therefore, the 1-dimensional model with a tile approach according to the footprint of each eddy-covariance measurement is the only realistic approach.

A further problem is the so-called unclosed energy balance. The energy balance with turbulence measurements (eddy-covariance method, Aubinet et al., 2012) is not closed by an amount of up to 30 % according to the overview paper by Foken (2008), while

the models close the energy balance by definition. The main reason for this is most likely the presence of fluxes caused by larger turbulence structures like secondary circulations (Foken, 2008). Measurement data must be corrected for energy balance closure before comparison with model outputs can be made. The missing flux will be distributed between the sensible and the latent heat flux according to the Bowen ratio (Twine et al., 2000) or the buoyancy flux (Charuchittipan et al., 2014). Carbon and other trace gas fluxes have not been corrected before the present study.

Simulating the turbulent transfer for heterogeneous landscapes utilizing a one-dimensional SVAT model (Soil-Vegetation-Atmosphere Transfer) represents a multi-faceted challenge. For forest, coherent structures are a typical phenomenon of turbulent exchange (Gao et al., 1989; Bergström and Högström, 1989), and they can be measured with the eddy covariance technique. However, coherent structures modify the turbulent transfer of energy and matter in a manner that complicates capturing them with models. Higher order closure models are able to overcome this problem (Deardorff, 1966). In this study the Advanced

Canopy-Atmosphere-Soil Algorithm (ACASA, Pyles et al., 2000) has been utilized with a third order closure scheme (Meyers and Paw U, 1986), and a more exact resolution of the turbulent structure for forest is provided. Thus, the ACASA model is much better suited to handling coherent structures and counter gradients than classical SVAT models with a first order closure (Staudt et al., 2011). Falge et al. (2017) provides a summary of the results of all model studies made in the last nearly 20 years at the Waldstein-Weidenbrunnen site, without any discussion of specific problems.

This paper is based on the earlier study by Falge et al. (2017), where the ACASA model was insufficiently applied for the clearing and compared with eddy-covariance data. As a first guess, the tile approach for the clearing was derived from the relative percentages of the individual land cover classes of the whole clearing without any selection according to the footprint. Furthermore, the measured fluxes were corrected for the energy balance closure. The present paper extends and completes the first paper in the following directions: (i) For the clearing, a more precise concept of the comparison of flux measurements with

1-D modeling using a tile approach according to the half-hourly footprints. (ii) For the forest, a detailed discussion of the two

different energy balance correction methods is provided. The inclusion of both surfaces in this study is necessary, because flux studies often use MODIS (Moderate-resolution Imaging Spectroradiometer) data with a grid size of 500 m or 250 m. Since the windthrow in 2007, the satellite sees about 50 % forest and 50 % clearing in the relevant grid.

## 2 Material and methods

### 2.1 Waldstein-Weidenbrunnen site

The experimental data for the initialization of the model and the evaluation of its outputs were collected during the third intensive observation period (IOP3) of the EGER (ExchanGE processes in mountainous Regions) project (Foken et al., 2012). The IOP3 campaign took place from 13 June to 26 July 2011. The EGER project mainly aimed at capturing the relevant exchange processes within the soil-vegetation-atmosphere framework and their interactions at different scales, and the work in IOP3 thereby focused on the role of surface heterogeneities in atmospheric exchange and chemistry as well as biogeochemistry.

The experimental site (50°8'N, 11°52'E, about 775 m a.s.l.) is part of the Bayreuth Centre of Ecological and Environmental Research (BayCEER) and lies in the upper part of the 'Lehstenbach' catchment (Fig. 1a). It is situated between two hilltops: to the south-west the 'Großer Waldstein' (879 m a.s.l.) and the 'Bergkopf' (857 m a.s.l.) to the north-east. The Lehstenbach catchment is part of the slopes of the 'Waldstein' in the north-western part of the 'Fichtelgebirge' mountains. The 'Fichtelgebirge' is a low mountain range in north-eastern Bavaria, Germany and is mostly densely forested.

The measurements of IOP3 were carried out in a spruce forest next to the FLUXNET site 'Waldstein-Weidenbrunnen' (DE-Bay) and in a nearby clearing with heterogeneous low vegetation located to the south of the FLUXNET site (Fig. 1b). The clearing was created by a wind throw on January 18, 2007 that was induced by the European windstorm Kyrill (Foken et al., 2012). The vegetation of the measurement site is heterogeneous and there is a slope of 3 ° from the pine forest to the clearing (from north to south). A more detailed description of the surrounding topography is provided by Foken et al. (2017a).

The forest consists mainly of Norway spruce (*Picea abies*) with a stand height of about 27 m and a leaf area index (LAI) of $4.8\,\mathrm{m^2\,m^{-2}}$ (Foken et al., 2017a). The stand age is circa 60 years (estimate from 2013) and the forest structure is characterized by an open trunk space as well as a dense crown space. Most needles are located between $0.5\,z\,h_\mathrm{c}^{-1}$ and $0.9\,z\,h_\mathrm{c}^{-1}$, where $z$ is the measurement height and $h_\mathrm{c}$ the stand height of 27 m. In the ground area, the understory comprises two-thirds crinkled hairgrass (*Deschampsia flexuosa*) and moss (together LAI $\leq 0.5\,\mathrm{m^2\,m^{-2}}$) and one third characterized by blueberry (*Vaccinium myrtillus*) and young Norway spruce (*Picea abies*, together PAI of $3.5\,\mathrm{m^2\,m^{-2}}$). For more details see Foken et al. (2012). The information regarding the overstory trees is summarized in Table 1. The vegetation of the clearing is a conglomeration of young spruce trees, blueberry (*Vaccinium myrtillus*), and different grasses (e. g. *Calamagrostis, Agrostis, Poaceae*). However, dead wood as well as bare soil are also elements of the clearing, and together they make up one third of the ground cover of this area. The clearing also contains approximately 2 % young deciduous trees: alder (*Alnus*), maple (*Acer*), beech (*Fagus*), and sorbus (*Sorbus*), with a greater canopy height than the remaining species. Due to the heterogeneous vegetation at the clearing, no distinct mean canopy height can be estimated for the clearing. Table 1 provides an overview of the species present and their mean canopy height.

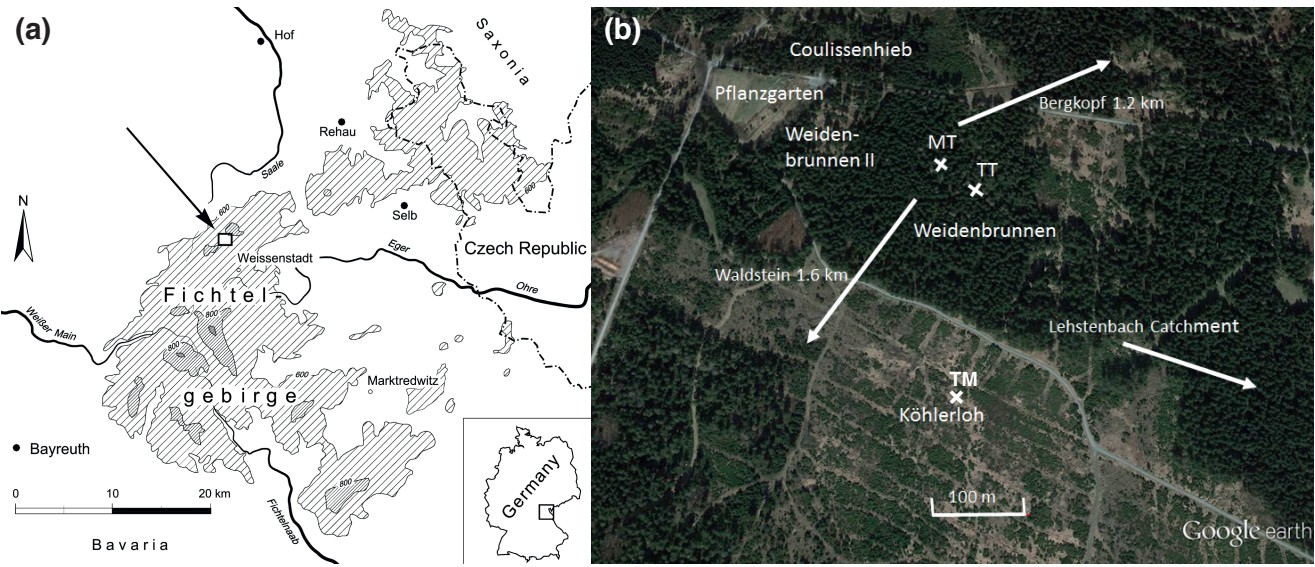

**Figure 1.** (a) Location of the BayCEER research site at the 'Waldstein' hillsides in the 'Fichtelgebirge' mountains (modified from Gerstberger et al. (2004)), published with kind permission of ©Springer, Berlin, Heidelberg 2004, All Rights Reserved; (b) Aerial picture of the patchy landscape showing meteorological towers/masts as well as the measuring points that are important for this study. MT: Main Tower, TT: Turbulence Tower, TM: Turbulence Mast (modified from Foken et al. (2017a), Published with kind permission of ©Springer, Berlin Heidelberg 2017 and ©Google earth, 2015, All Rights Reserved).

## 2.2 Experimental setup and data

During IOP3, high-frequency turbulence measurements were conducted at different measurement towers (Fig. 1b). For this study, especially the data from the towers in the forest (MT, TT) and the turbulence mast at the clearing (TM) have been utilized. The towers and masts were generally equipped with 3D sonic anemometers, to enable collection of information about
wind components ($u$, $v$, $w$) and the sonic temperature ($T_S$). High-frequency gas analyzers for carbon dioxide ($c_{CO_2}$) and water vapor ($q$) were installed in conjunction with sonic anemometers. This allowed the turbulent exchange for forest and clearing to be investigated with the devices as summarized in Table 2, and this has been utilized for comparison with simulations. Raw flux data (20 Hz) have been processed with the TK3 software (Mauder and Foken, 2015). The flux data have been additionally classified by their quality according to the classification scheme by Foken et al. (2004). For comparison with model results,
only data with quality flags of 6 and better have been utilized. Furthermore, the flux measurements have been energy balance corrected with the methods described in Sect. 2.4.

The ACASA model needs half-hourly meteorological input values as well as mean values of soil temperature and soil moisture for initialization. These input values (see Table 2) are provided by standard measurements at the "Pflanzgarten" (Fig. 1b) and measurements at the Main Tower (MT) for the spruce forest as well as at the Turbulence Mast (TM) for the clearing. Table 2
provides an overview of the meteorological input data for the two simulated sites and the corresponding instrumentation.

**Table 1.** Vegetation at the forest and the Köhlerloh clearing with their main characteristics according to Foken et al. (2017a).

| Species | Ground cover in % | Height in m | PAI in $\mathrm{m^2\,m^{-2}}$ | MSC[†] in $\mathrm{mol\,m^{-2}\,s^{-1}}$ |
|---|---|---|---|---|
| **Forest** | | | | |
| Spruce (*Picea abies*) | 100 | 27 | $5.6 \pm 2.1$ | 0.001 |
| **Clearing** | | | | |
| Crinkled hairgrass (*Deschampsia flexuosa*) | 21.7 | $0.17 \pm 0.05$ | $2.65 \pm 1.08$ | 0.127 |
| Young spruce (*Picea abies*) | 21.4 | $1.21 \pm 0.50$ | $8.67 \pm 2,29$ | 0.054 |
| Blueberry (*Vaccinium myrtillus*) | 15.9 | $0.27 \pm 0.10$ | $3.46 \pm 1.05$ | 0.050 |
| Reed, bent, and true grasses (*Calamagrostis, Agrostis, Poaceae*) | 9.0 | $0.42 \pm 0.11$ | $3.43 \pm 1.07$ | 0.049 |
| Rush and sedge family (*Juncaceae, Cyperaceae*) | 3.1 | $0.74 \pm 0.13$ | $1.77 \pm 0.60$ | 0.002 |
| Other herbaceous[*] | 1.6 | / | / | 0.002 |
| Moss | 0.9 | / | / | / |
| Dead grass, bare soil | 7.2 | / | / | / |
| Dead wood | 18.8 | / | / | / |
| Small ditches | 0.2 | / | / | / |

[†] MSC stands for minimal stomata conductance and is a model parameter utilizied in ACASA

[*] This includes: *Digitalis purpurea* (common foxglove), *Epilobium angustifolium* (willow herb), and *Urtica dioica* (stinging nettle)

Three Golden Day Periods (GDPs) were selected within IOP3 because of fair weather conditions and predominantly good performance of the measurement devices. The first GDP was from 26 to 29 June 2011 (corresponding to the 177 to 180 day of the year, DOY). On the last three days of this period, clear sky conditions prevailed until 2 p.m. as well as moderate westerly (26./ 27.) or easterly (28./ 29.) winds. The second GDP occurred from 4 to 8 July 2011 (DOY: 185 to 189), with the best weather conditions on 6 and 7 July and partly overcast sky on the rest of the days. The wind in this period was weak and blew from the west (4./7./8.) or east (5./6.). Minor amounts of precipitation were measured on 4 and 8 July. For the last GDP from 14 to 17 July 2011 (DOY: 195 to 198) data are missing for the clearing at 5.5 m.

## 2.3 The ACASA model

ACASA (Pyles et al., 2000) was utilized to simulate turbulent transfer of heat, water vapor, and $CO_2$ for spruce forest and clearing. ACASA is a multi-layer SVAT model developed at the University of California, Davis. The main feature of ACASA is that the turbulent transfer within and above the canopy is calculated by a diabatic, third-order closure method, which is based on the theoretical work of Meyers and Paw U (1986, 1987). The multi-layer structure of ACASA is represented by 20 evenly distributed atmospheric layers, which reach from trunk space to twice the canopy height, and by 15 soil layers. For

**Table 2.** Meteorological instrumentation for the determination of the sensible and the latent heat flux as well as the net ecosystem exchange (NEE) for forest and clearing at different measurement towers in distinct heights, using the eddy-covariance technique (with the frequency – 20 Hz – applied as indicated), and additionally, the most important input parameter of the model (for this purpose the mean $CO_2$ concentrations have been averaged from the turbulence data). Documentation of the complete data set is available in Foken (2017a).

| Measurement site | Parameter | Height | Device |
|---|---|---|---|
| **Pflanzgarten** | Precipitation | 1 m | OMC 212, Adolf Thies GmbH & Co. KG |
| | Pressure | 2 m | AB 60, Ammonit Gesellschaft für Messtechnik mbH |
| **Weidenbrunnen, Main Tower (MT)** | Wind vector and sonic temperature (20 Hz) | 32.5 m | USA-1, METEK GmbH |
| | Humidity (20 Hz) | 32 m | LI-7000[†], LI-COR Biosciences |
| | Temperature and relative humidity | 31 m | Frankenberger ventilated psychrometer, Theodor Friedrichs & Co |
| | Wind velocity | 31 m | Cup anemometer, Theodor Friedrichs & Co |
| | Up- and downwelling shortwave radiation | 29.5 m | CM 14, Kipp & Zonen |
| | Up- and downwelling longwave radiation | 29.5 m | CG 2, Kipp & Zonen |
| | Soil heat flux and temperature profile | -0.08 m | HFT, Campbell Sci. Inc. |
| **Weidenbrunnen, Turbulence Tower (TT)** | $CO_2$ concentration (20 Hz) | 36 m | LI-7500 [§], LI-COR Biosciences |
| | Wind vector (20 Hz) | 36 m | USA-1, METEK |
| **Köhlerloh Turbulence Mast (TM)** | Wind vector and sonic temperature (20 Hz) | 2.25 m/5.5 m | CSAT3, Campbell Sci. Inc. |
| | Humidity (20 Hz) | 2.25 m/5.5 m | LI-7000[†] / LI-7500 [§], LI-COR Biosciences[*] |
| | $CO_2$ concentration (20 Hz) | 2.25 m/5.5 m | LI-7000[†] / LI-7500 [§], LI-COR Biosciences[*] |
| | Up- and downwelling shortwave radiation | 2.0 m | CNR4, Kipp & Zonen |
| | Up- and downwelling longwave radiation | 2.0 m | CNR4, Kipp & Zonen |
| | Soil heat flux and temperature profile | -0.15 m | HP3, Rimco |

[*] LI – 7000 in 2.25 m and LI – 7500 in 5.5 m.

[†] closed path gas analyzer, [§] open path gas analyzer.

the calculation of leaf, stem, and soil surface temperatures, the fourth-order polynomial scheme of Paw U and Gao (1988) is incorporated. Utilizing this polynomial allows the calculation of the temperatures of these components without making substantial errors in the case of significant deviations from the ambient temperature. The model was considerably updated by the University of Davis (Pyles et al., 2000, 2003) and it is still in use (Falk et al., 2014). For this study, the ACASA model was adapted from a version of April 2013.

Direct as well as diffuse radiation can be absorbed, transmitted, or reflected by the canopy, whereby these processes are dependent on the leaf and branch distribution of the plants. For this purpose, the above ground biomass is distributed in 10

different leaf angle classes (Meyers, 1985). In order to enable a preferably exact allocation of the energy to sensible and latent heat fluxes, a leaf energy balance equation has to be solved. Plant physiological feedback to micro-environmental conditions is incorporated by approaches of Leuning (1990) and Collatz et al. (1991) to the Ball-Berry stomatal conductance combined with photosynthetic rates calculated after an equation of Farquhar and Caemmerer (1982), whereby coupling follows standards of Su and Paw U (1996). ACASA also includes canopy heat storage, which is mainly important for tall vegetation, and canopy interception of precipitation. For thermal and hydrological aspects the soil model is based on the one-dimensional diffusion equation (de Vries, 1952) as incorporated in MAPS (Mesoscale Analysis and Prediction System, Smirnova et al., 1997, 2000). About ten years ago the University of Bayreuth started to work with ACASA. The first issue was to use a sensitivity analysis to check whether the model could be applied for a Central European spruce forest (Staudt et al., 2010). We also found some model-specific problems, and the model was again updated by the University of Davis. Using the updated model, a first study for the Waldstein-Weidenbrunnen site was published (Staudt et al., 2011). This study showed that the ACASA model is more accurate (in comparison to a K-approach model) when coherent structures dominate at night, which has been investigated in many experiments and summarized by Thomas et al. (2017). The model was changed by Staudt et al. (2011) for the spruce-specific parameterizations in three parts. The first change relates to the soil respiration calculations and had already been examined by Staudt et al. (2010) for spruce. The second change was made in the calculation of photosynthesis according to Falge et al. (1996) because temperatures less than $10\,^\circ$C are very common at the experimental site. The third adjustment relates to the day respiration for leaves because Tcherkez et al. (2008) quantified the day respiration as about one half of the night respiration rate. Furthermore, Falge et al. (2017) determined several parameters of the plants of the clearing (see Table 16.1 of Falge et al., 2017).

## 2.4 Energy balance correction

The eddy-covariance data of sensible heat, latent heat, and carbon dioxide flux have been energy balance corrected for the comparison with the simulations.

For the correction of the energy fluxes, the residual (Res) arises from the following equation:

$$\text{Res} = Q_\text{S} - H - LE - G - \Delta S, \tag{1}$$

where, $Q_\text{S}$ denotes net radiation, $H$ sensible heat flux, $LE$ latent heat flux, and $\Delta S$ the heat storage term of the above ground biomass, which was determined according to the investigations of Haverd et al. (2007) and Lindroth et al. (2010). Here, the heat storage term was only estimated for the spruce forest because of its larger biomass compared to the vegetation in the clearing. The heat storage was calculated with the following relation:

$$\Delta S = m_\text{f} c_\text{f} \frac{(T_\text{a}(t) - T_\text{a}(t - \Delta t))}{\Delta t}, \tag{2}$$

with $m_\text{f}$ the above ground biomass of the forest, $c_\text{f}$ the specific heat capacity of wood ($c_\text{f} = 1702.8\,\text{J}\,\text{kg}^{-1}\,\text{K}^{-1}$), and $T_\text{a}$ the air temperature in the spruce canopy. In this study, two different methods for energy balance correction have been applied to the measurements of IOP3. The first correction method is introduced by Twine et al. (2000) and preserves the Bowen ratio.

This method is usually utilized for the correction of heat fluxes under the assumption of measuring errors, but requires their scalar similarity. Henceforth, this method is abbreviated with EBC-Bo. The second correction method accounts for near surface secondary circulations, which transport more sensible heat. Therefore, the alternative correction method (EBC-HB) proposed by Charuchittipan et al. (2014) utilizes the buoyancy flux ratio. Here, for the typical range of the Bowen ratio, more residual

energy is ascribed to the sensible heat flux, as was done in several studies (Foken et al., 2011). The difference to the EBC-Bo correction is especially pronounced for low Bowen ratios. At night, the applicability of the corrections is limited due to the occurrence of minor or negative heat fluxes.

The discrepancy between measured and simulated NEE can be an effect of the unclosed energy balance on the $CO_2$ fluxes. Therefore, we propose a new correction method on the basis of the good scalar similarity between the humidity and the carbon

dioxide concentration mainly in the first part of the day, for scalar similarity see Ruppert et al. (2006), while the similarity between the temperature and other scalars often fails (Pearson et al., 1998). The measured NEE is corrected by referring to the energy balance corrected latent heat flux, which is introduced in Falge et al. (2017). For this purpose, the ratio $k$ between the corrected and the uncorrected latent heat fluxes is calculated for both correction methods:

$$k_{\mathrm{Bo}} = \frac{LE_{\mathrm{Bo}}}{LE} \ \text{ and } \ k_{\mathrm{HB}} = \frac{LE_{\mathrm{HB}}}{LE} \ . \tag{3}$$

The measured NEE value was then multiplied with the respective correction factor $k_{\mathrm{Bo}}$ or $k_{\mathrm{HB}}$. This correction method is proposed for the NEE for the first time and the applicability of this approach is therefore evaluated. The correction is not very large. Therefore, until shown otherwise, the energy balance correction can be neglected for scalar fluxes.

## 2.5   Footprint model

Nowadays, footprint models are a commonly used tool for the identification of the source area of flux measurements (Leclerc

and Foken, 2014). For the footprint calculations of this study, a Lagrangian footprint model is applied (Rannik et al., 2000, 2003), whereby a spectral method of the non-linear flux averaging of surface characteristics (friction, roughness length) according to Hasager and Jensen (1999) is employed (Göckede et al., 2006) because the linear averaging of the roughness length (input parameter of footprint models) is inaccurate. The sensitivity of the Lagrangian footprint model to the turbulence statistics was tested by Göckede et al. (2007) for the Waldstein-Weidenbrunnen site. A footprint climatology and footprint studies,

including the effect of the footprint on the data quality, are available from Foken et al. (2017b). On average, more than 80 % of the target area is forest. Only for southerly winds and stable stratification does the Köhlerloh clearing have a significant influence on the fluxes measured at the Main Tower MT.

In a study by Reithmaier et al. (2006) the land use for the Waldstein-Weidenbrunnen site was determined and mapped, and this was tested for footprint applications. Due to the distinct small-scale heterogeneities, the land cover map applied here has a high

resolution (4.5 x 4.5 m$^2$). This grid size is in agreement with the size of the typical heterogeneities and the recommendations for footprint analyses for low measuring heights (Leclerc and Foken, 2014). The calculated footprints for thirty-minute time periods have been used to generate tile approaches of the model.

## 3  Results and discussion

### 3.1  Footprint climatology for the Köhlerloh clearing

The footprint climatology of the mainly forested Waldstein-Weidenbrunnen site is well described in many publications like Göckede et al. (2008) and summarized by Foken et al. (2017b). In the following, the footprint climatology of the clearing site Köhlerloh (Turbulence mast TM) is analyzed. Only for southerly winds and stable stratification does the Köhlerloh clearing have a significant influence on the fluxes measured at the Main Tower (MT). Therefore, only the footprint of the Köhlerloh site (Turbulence mast TM) is analyzed in the following.

As an overview, the footprint climatology for the period from June, 26 to July, 17 2011 has been calculated, and is a superposition of all individual footprints. Fig. 2 shows the footprint climatology for the Turbulence Mast TM in two different heights at the clearing. The main difference in the footprint climatology between the different stratifications appears for stable conditions in both measuring heights. For unstable and neutral stratification, mainly the clearing contributes to the turbulent fluxes at the Turbulence Mast TM. For stable stratification and northerly wind directions, the spruce forest has additional influence on the measured turbulent flux at TM (see Fig. 2). For 5.5 m measuring height, the footprint source areas are generally enhanced/extended. Regarding the stable stratification case, the north-westerly part of the spruce forest is only included in the outer source region of the turbulent for flux 2.25 m measuring height and is more pronounced for 5.5 m measuring height. Additionally, a slight influence from the north-easterly spruce areas connected to the forest edge is calculated for 5.5 m measuring height.

### 3.2  Comparison of modeled and measured fluxes

#### 3.2.1  Daily cycles

Falge et al. (2017) have already compared the sensible and the latent heat flux as well as the NEE for the forest (MT) and the clearing (TM, 2.25 m). They found that the sensible heat flux corrected with the buoyancy flux (EBC-HB) agrees better with the modeled data than the flux corrected with the Bowen ratio (EBC-Bo) for the first Golden Day Period. The result for the latent heat flux is similar and the NEE shows no significant differences. The NEE over the clearing was only about 50 % of the value over the forest.

In the following, we compare for the clearing the measured as well as the energy balance corrected fluxes, with the tile approach of the model according to i) the relevant footprint near the measurement point and ii) the land cover distribution of the whole clearing. It was found that for the footprint of 2.25 m height there are no significant differences between either modeling approach due to the small size of the footprint (not shown). The results for TM with the footprint for 5.5 m measuring height are shown in Fig. 3 for the first and in Fig. 4 for the second Golden Day Period. The difference between both model approaches is more significant in the first period, which had a Bowen ratio $Bo \geq 1$, while in the second period the Bowen ratio was often below 1. The difference in the land use characteristics were more dominant under drier conditions.

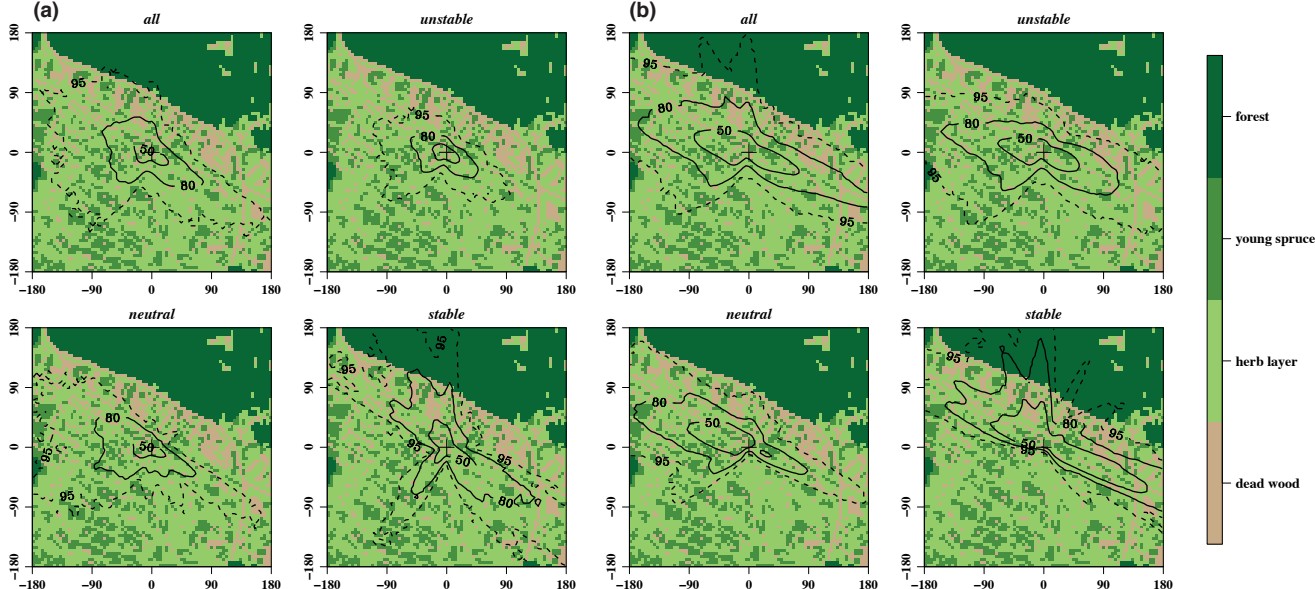

**Figure 2.** Footprint climatology of the Köhlerloh clearing (26.06. to 17.07.2011) at the Turbulence Mast TM for the three stability classes and the combination of all classes; a) Left four panels at 2.25 m measuring height; b) Right four panels at 5.5 m measuring height. Here, four different land-use classes are considered, with the herb layer being composed of *Deschampsia*, *Calamagrostis*, *Juncus*, and *Vaccinium*. Distances are given in m.

From the visual comparison of Figs. 3 and 4 it has been found that the buoyancy corrected sensible and latent heat fluxes obviously better agree with the model than do the Bowen ratio corrected fluxes. The scatter is too large to allow any trend for the NEE to be seen. There is also no clear finding as to whether the footprint weighted tile approach of the model (non-weighted approach not shown) gives better results. The integrated fluxes of the tile approaches for the whole clearing and the measured fluxes within the footprint of the turbulence mast for 5.5 m height do not differ significantly, except for stable atmospheric stratification (cf. Fig. 2). Probably not only the land cover characteristics, but also the soil water content have a significant influence. This can be seen by the high measured latent heat fluxes in the first Golden Day Period (see Fig. 3). Regression analyses are discussed in the following section for both weighting methods.

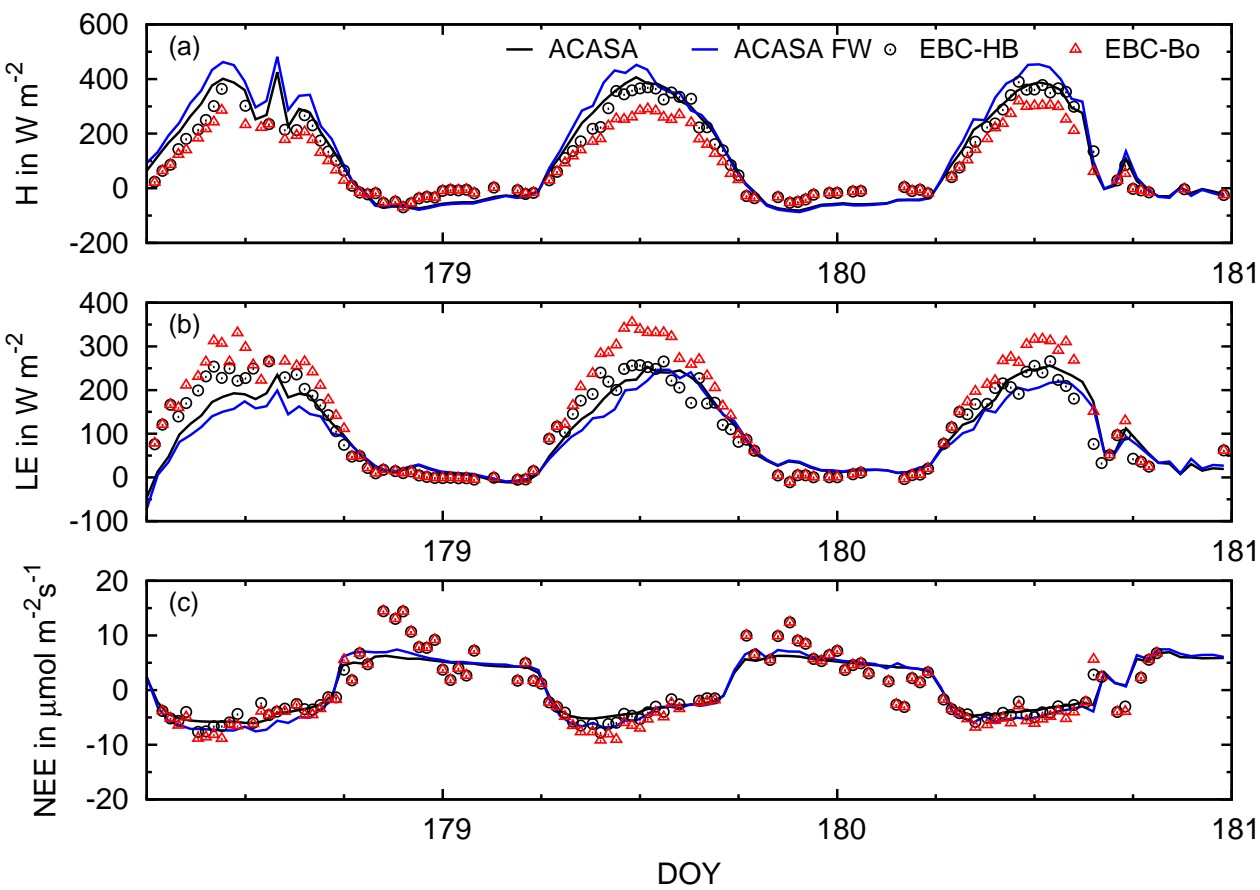

**Figure 3.** Comparison of ACASA model simulations with measured turbulent fluxes at TM (in 5.5 m measurement height) for the first GDP; a) Sensible heat (H); b) Latent heat (LE) and c) net ecosystem exchange (NEE). The black solid line displays ACASA simulations weighted by the mean plant distribution of the whole clearing and the blue solid line indicates ACASA simulations weighted by different plant classes within the footprint of the flux measurements. Energy balance corrected flux measurements are shown by black circles for the correction utilizing the buoyancy flux ratio (EBC-HB) and red triangles for correction with the Bowen ratio (EBC-Bo).

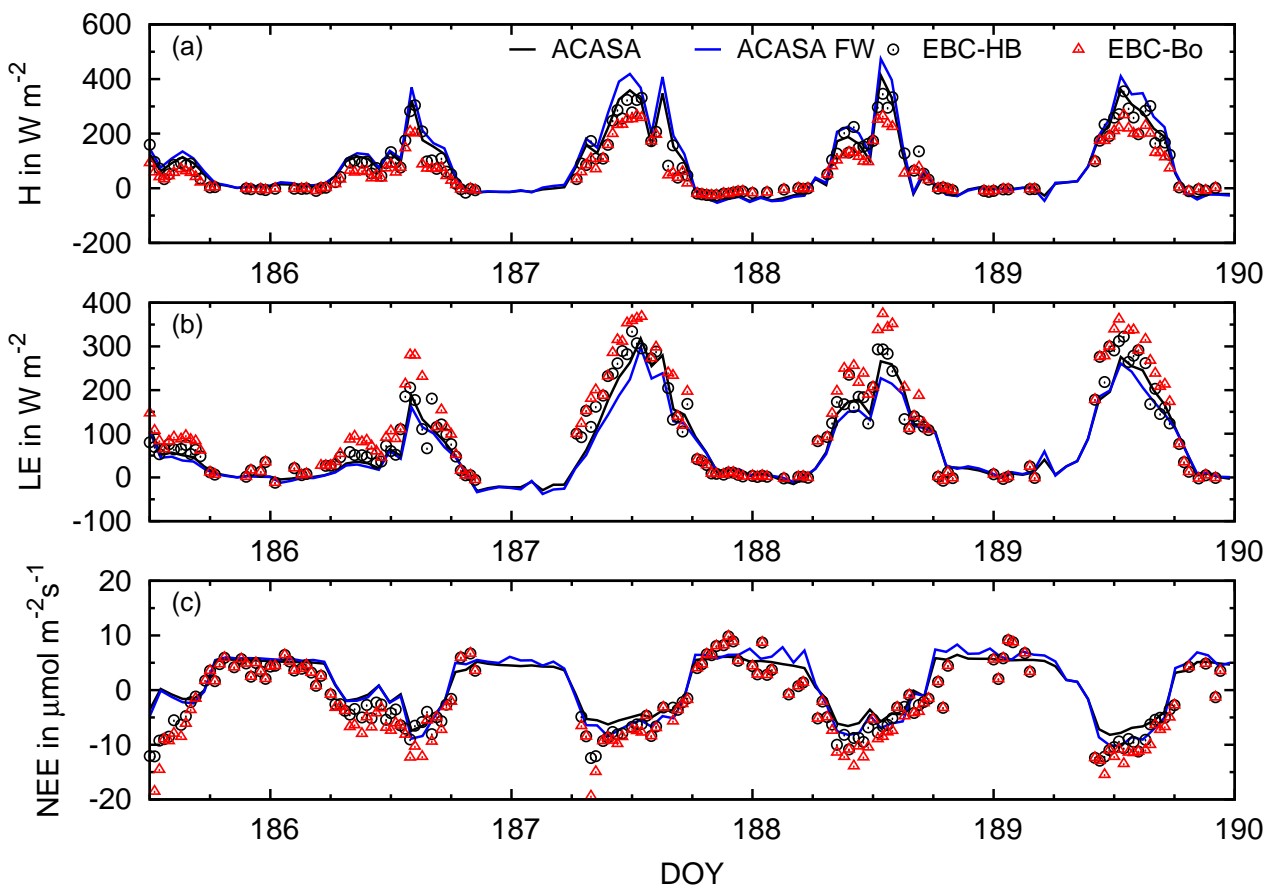

**Figure 4.** Same as Fig. 3 for the second GDP.

### 3.2.2 Regression analyses

Neither of the modeled nor the measured fluxes are free of errors and could be used as an independent parameter. Therefore, the errors are assumed to be similar and an orthogonal regression analysis for evenly distributed errors has been applied (Dunn, 2004). The regression has been done for the first two Golden Day Periods for the forest and the clearing (with and without footprint weighted tile approach) for the sensible (H) and the latent (LE) heat flux as well as the net ecosystem exchange (NEE). Furthermore, the available energy (AE), as the difference between net radiation and the ground heat flux (Liebethal et al., 2005) as well as the heat storage in the biomass (the latter is only determined for forest), is shown. The results are given in Tables 3 and 4 as well as being partly illustrated in Fig. 5. Due to a non-equal distribution of the data points, the data have been binned for flux classes of the modeled data, whereby the number of bins have been chosen so that the measured flux data

is equally distributed. Therefore, the observed trend in the data have not been artificially shifted due to the applied binning and further, the excluded data (for cases with $Bo < 0$ and too small nighttime fluxes) have merely been located in one bin. Fig. 5 is created without the confidence intervals for the regression curves because the fluxes only marginally scatter around the average regression curve and for the sake of improved clarity. Tables 3 and 4 additionally contain the results for the regressions through the point of origin because of the large intercepts observed in regression analysis. Nevertheless, the slope did not vary substantially between both regression methods.

The analysis of the eddy-covariance data show that in all cases, except for the latent heat in the second Golden Day Period, the fluxes are underestimated and an energy balance correction is necessary to obtain an agreement with the modeled data. All results are very similar for both Golden Day periods. For a discussion of the results included in Fig. 5 as well as Tables 3 and 4, the available energy is also shown and the therein considered fluxes are summarized. The net radiation is underestimated by the model by about $5-7\%$ for both sites (not shown). The ground heat flux and the storage in the biomass both amount to less than $10\%$ of the net radiation, with a large scatter between the measured and the modeled data. A significant difference from the 1:1 relationship has only been found for the ground heat flux of the clearing, which is higher by a factor of 2 for the experimental data. Due to the installation of the sensors the ground was probably less covered with vegetation, and thus could store more energy. For the forest site, the underestimation of the available energy by the model was only about $4\%$. For the clearing site, the modeled available energy has a constant offset of $30\,\mathrm{W\,m^{-2}}$ and was therefore higher than the experimental data.

The effect of the different energy balance corrections is not equal for both sites. For the forest site, both corrections of the sensible heat flux agree quite well with the model, with slightly better values for the buoyancy correction. For the latent heat flux of the forest site, the measurements corrected according to the Bowen ratio are in better agreement with the simulations. For $Bo > 1$, the buoyancy correction, which assumes that only convection is the reason of the unclosed energy balance, overestimates the energy balance closure correction and the true correction might lie between both correction methods. The turbulent fluxes might be slightly higher than the modeled fluxes because of the measured available energy being about $4\%$ higher, as can be seen after applying both corrections (see also Sect. 3.2.3).

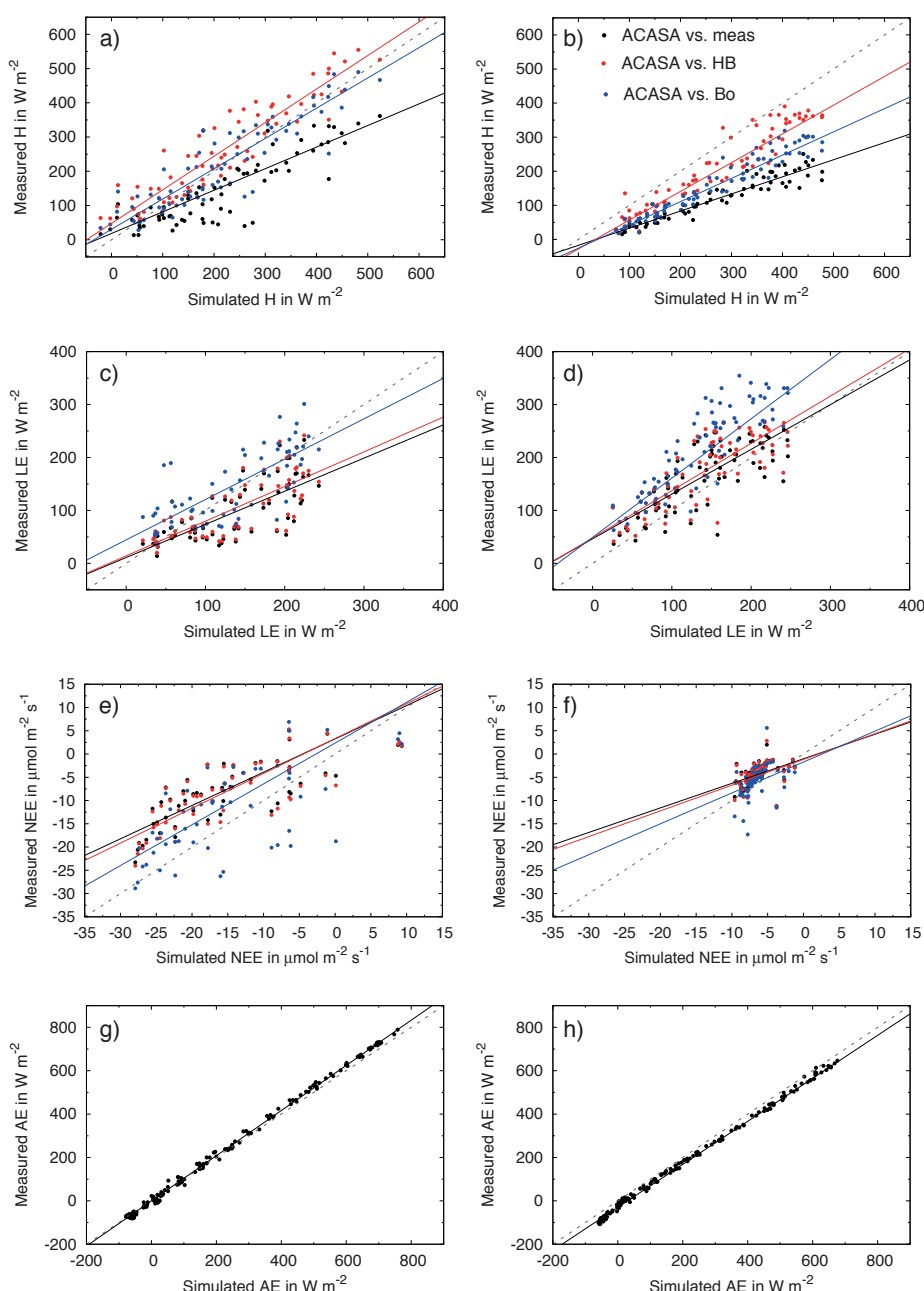

**Figure 5.** Example of the regression analysis of the measured and the simulated (ACASA) data for GDP 1 for a) and b) the sensible heat flux (H); c) and d) the latent heat flux (LE); e) and f) the net ecosystem exchange (NEE); and g) and h) the available energy (AE) above the forest (a, c, e, g) and above the clearing at 5.5 m height (footprint weighted modeled fluxes; b, d, f, h) for uncorrected (black), Bowen ratio corrected (blue), and buoyancy corrected (red) eddy-covariance data. The regression lines are based on binned data for $50\,\mathrm{W\,m^{-2}}$ or $2/4\,\mu\mathrm{mol\,m^{-2}\,s^{-1}}$ classes of the modeled data and an orthogonal regression. The analysis has been achieved for the first and second Golden Day Periods, with Tables 3 and 4 displaying the results of an orthogonal regression with as well as without assuming a zero intercept for the clearing and the forest, respectively.

**Table 3.** Results of the comparison of the measured and the modeled flux data for different correction methods with an orthogonal regression for binned data points for the clearing with and without footprint weighting (for details see Fig.5, absolute values in W m$^{-2}$ or $\mu$mol m$^{-2}$ s$^{-1}$ for the NEE).

| Flux correction | First Golden Day Period, June 26-28 | | | | | Second Golden Day Period, July 4-8 | | | | |
|---|---|---|---|---|---|---|---|---|---|---|
| | Slope | Absolute value | Slope zero intercept[§] | Corr. coeff. | Data bins | Slope | Absolute value | Slope zero intercept[§] | Corr. coeff. | Data bins |
| **Sensible heat flux, clearing 5.5 m, footprint weighted tile approach** | | | | | | | | | | |
| No | 0.50 | -17.3 | 0.45 | 0.99 | 9 | 0.52 | -15.8 | 0.46 | 0.98 | 9 |
| Buoyancy | 0.84 | -27.9 | 0.76 | 0.99 | 9 | 0.72 | 9.3 | 0.75 | 0.99 | 9 |
| Bowen | 0.68 | -25.7 | 0.61 | 1.00 | 9 | 0.61 | -8.0 | 0.59 | 0.99 | 9 |
| **Sensible heat flux, clearing 5.5 m, non-footprint weighted tile approach** | | | | | | | | | | |
| no | 0.61 | -25.4 | 0.51 | 1.00 | 7 | 0.59 | -22.7 | 0.51 | 0.98 | 8 |
| buoyancy | 1.01 | -34.9 | 0.77 | 0.99 | 7 | 0.84 | 8.3 | 0.87 | 1.00 | 8 |
| Bowen | 0.83 | -36.1 | 0.70 | 1.00 | 7 | 0.71 | -11.7 | 0.67 | 0.99 | 8 |
| **Latent heat flux, clearing 5.5 m, footprint weighted tile approach** | | | | | | | | | | |
| No | 0.84 | 46.4 | 1.14 | 0.98 | 5 | 1.02 | 13.7 | 1.09 | 0.98 | 6 |
| Buoyancy | 0.89 | 48.1 | 1.20 | 0.98 | 5 | 1.05 | 20.4 | 1.16 | 0.99 | 6 |
| Bowen | 1.20 | 49.1 | 1.51 | 1.00 | 5 | 1.25 | 36.5 | 1.44 | 0.99 | 6 |
| **Latent heat flux, clearing 5.5 m, non-footprint weighted tile approach** | | | | | | | | | | |
| no | 0.84 | 36.4 | 1.07 | 0.97 | 6 | 1.03 | -5.3 | 1.00 | 0.98 | 6 |
| buoyancy | 0.92 | 34.0 | 1.14 | 0.97 | 6 | 1.05 | 1.8 | 1.06 | 0.98 | 6 |
| Bowen | 1.19 | 39.2 | 1.43 | 0.99 | 6 | 1.16 | 29.5 | 1.31 | 0.99 | 6 |
| **NEE, clearing 5.5 m, footprint weighted tile approach** | | | | | | | | | | |
| No | 0.53 | -1.0 | 0.69 | 0.88 | 5 | 0.52 | -2.3 | 0.84 | 0.94 | 7 |
| Buoyancy | 0.55 | -1.2 | 0.74 | 0.87 | 5 | 0.55 | -2.6 | 0.90 | 0.94 | 7 |
| Bowen | 0.66 | -1.7 | 0.94 | 0.86 | 5 | 0.66 | -3.9 | 1.20 | 0.94 | 7 |
| **NEE, clearing 5.5 m, non-footprint weighted tile approach** | | | | | | | | | | |
| no | 0.64 | -1.9 | 1.04 | 0.98 | 5 | 0.74 | -3.0 | 1.25 | 0.98 | 6 |
| buoyancy | 0.62 | -2.1 | 1.08 | 0.98 | 5 | 0.76 | -3.3 | 1.33 | 0.98 | 6 |
| Bowen | 0.55 | -3.1 | 1.25 | 0.97 | 5 | 0.89 | -4.7 | 1.71 | 0.97 | 6 |
| **AE, clearing 5.5 m, footprint weighted tile approach** | | | | | | | | | | |
| Uniform[†] | 0.99 | -30.0 | 0.93 | 1.00 | 16 | 0.97 | -18.2 | 0.93 | 1.00 | 15 |
| **AE[†], clearing 5.5 m, non-footprint weighted tile approach** | | | | | | | | | | |
| uniform | 1.04 | -31.3 | 0.97 | 0.97 | 15 | 1.05 | -23.7 | 1.00 | 1.00 | 15 |

[§] Orthogonal regression considering a zero intercept; [†] AE is the same for all 3 methods because the heat storage term of the canopy and the ground heat flux are uniformly estimated

**Table 4.** Results of the comparison of the measured and the modeled flux data for different correction methods with an orthogonal regression for binned data points for the forest (absolute values in $\mathrm{W\,m^{-2}}$ or $\mu\mathrm{mol\,m^{-2}\,s^{-1}}$ for the NEE).

| Flux correction | First Golden Day Period, June 26-28 | | | | | Second Golden Day Period, July 4-8 | | | | |
|---|---|---|---|---|---|---|---|---|---|---|
| | Slope | Absolute value | Slope zero intercept[§] | Corr. coeff. | Data bins | Slope | Absolute value | Slope zero intercept[§] | Corr. coeff. | Data bins |
| **Sensible heat flux, forest 32 m** | | | | | | | | | | |
| No | 0.63 | 18.4 | 0.69 | 0.98 | 11 | 0.69 | -25.7 | 0.61 | 0.95 | 10 |
| Buoyancy | 0.98 | 47.8 | 1.13 | 0.99 | 11 | 0.91 | 25.6 | 0.99 | 0.99 | 10 |
| Bowen | 0.88 | 30.6 | 0.98 | 0.99 | 11 | 0.82 | 0.6 | 0.82 | 0.98 | 10 |
| **Latent heat flux, forest 32 m** | | | | | | | | | | |
| No | 0.63 | 11.0 | 0.69 | 0.95 | 5 | 0.67 | 23.8 | 0.79 | 0.99 | 6 |
| Buoyancy | 0.65 | 14.2 | 0.74 | 0.94 | 5 | 0.69 | 33.2 | 0.86 | 0.99 | 6 |
| Bowen | 0.76 | 44.4 | 1.04 | 0.91 | 5 | 0.82 | 58.5 | 1.12 | 0.99 | 6 |
| **NEE, forest 32 m** | | | | | | | | | | |
| No | 0.72 | 3.3 | 0.55 | 0.93 | 6 | 0.65 | -1.1 | 0.70 | 0.96 | 8 |
| Buoyancy | 0.75 | 3.3 | 0.58 | 0.93 | 6 | 0.64 | -2.4 | 0.76 | 0.95 | 8 |
| Bowen | 0.88 | 2.4 | 0.76 | 0.95 | 6 | 0.70 | -7.3 | 1.06 | 0.81 | 8 |
| **AE, forest 32 m** | | | | | | | | | | |
| Uniform[†] | 1.04 | -0.2 | 1.04 | 1.00 | 17 | 1.05 | -11.5 | 1.03 | 1.00 | 17 |

[§] Orthogonal regression considering a zero intercept; [†] AE is the same for all 3 methods because the heat storage term of the canopy and the ground heat flux are uniformly estimated

However, for the clearing with $Bo < 1$, different results have been found: with the Bowen ratio correction, the sensible heat flux is significantly underestimated by about 30 % and the latent heat flux is overestimated by about 12 %. The buoyancy correction underestimates both fluxes by about 10 %. The measured latent heat flux is often larger than the modeled latent heat flux. This is probably a large effect of the plants in the footprint and the soil water content, which is not proportional to the land cover fraction used by the footprint model. Taking into account the 30 $\mathrm{W\,m^{-2}}$ higher available energy of the model, which is in the order of about 10 – 20 % of the turbulent fluxes (therefore, the modeled fluxes are higher by this value), the buoyancy correction is more appropriate for the clearing.

The correction of the NEE data seems to be necessary, but the underestimation by the model is still given. This could be an overestimation by the measured fluxes due to the increased mechanical turbulence and consequently also turbulent fluxes caused by the heterogeneous forest structure, discussed by Foken (2017b). Furthermore, a related LES study, Kanani-Sühring and Raasch (2015) shows a maximal flux at a distance from the roughness step of about 10 times the canopy height that is

nearly the location of the tower. Therefore, the Bowen ratio correction of the NEE for the forest seems to agree better, while over the clearing the fluxes are too small to allow useful conclusions to be made.

### 3.2.3 Analysis of the energy balance closure correction

In this section we used the modeled data, which close the energy balance by definition – as an etalon, for the validation of the correction methods. Therefore, the modeled Bowen ratio was assumed to be true and the measured fluxes have been corrected in such a way that the energy balance was closed and the Bowen ratio agreed with the modeled Bowen ratio. Finally, the fraction of the residual attributed to the sensible heat flux has been determined and is shown in Fig. 6. Sometimes, when the measured latent heat flux was significantly larger than the modeled flux, the fraction was up to 150 %. This analysis was only made for the forest with a more homogeneous footprint. Figure 6 also shows, for comparison, the proposed corrections according to the Bowen ratio (Twine et al., 2000) and the buoyancy flux (Charuchittipan et al., 2014). Obviously, the assumption that convection is responsible for the energy balance problem and that a large part of the residual should be added to the sensible heat flux, may be only true for $Bo > 1.5$. For lower Bowen ratios, the real correction might lie between both methods, and for $Bo < 1$ be more in accordance with the Bowen ratio correction. Unfortunately, the number of the available data points for $Bo < 1.5$ is very low, so that the presented results can only be interpreted as a tendency. This means that secondary circulations, probably responsible for the unclosed energy balance, are not only generated by convective processes. Further research is necessary, including other well parametrized models that close the energy balance very well, and data sets for lower Bowen ratios. In all cases with high Bowen ratios, attribution of residual energy to latent heat flux is low. Therefore, due to the assumption of a similarity between the water and carbon dioxide fluxes, the NEE flux was only marginally corrected under these conditions. In conclusion, for spruce forests with typically high Bowen ratios, no correction of the NEE is necessary.

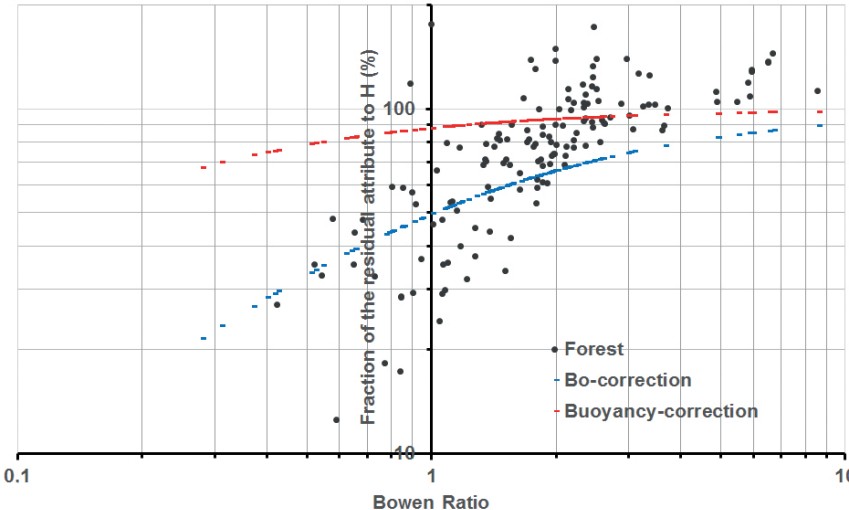

**Figure 6.** Fraction of the residual attributed to the sensible heat flux for the forest site (GDP 1, 2, and 3), under the assumption that the model calculated the true Bowen ratio, and according to the correction methods with the Bowen ratio (Twine et al., 2000) and the buoyancy flux (Charuchittipan et al., 2014).

### 3.2.4 Tile approach for a mixed forest site

The comparison of the modeled and the measured fluxes has shown that the ACASA model determines the fluxes for high and low vegetation with high accuracy, within the typical measurement uncertainty (Mauder et al., 2006, 2013) of eddy-covariance measurements, if the measured data are corrected for energy balance closure. This offers the possibility of modeling fluxes over
larger heterogeneous areas, like a catchment, with a tile approach. Even when the fluxes above the different land use types are significantly different and are changing, e.g.with the Bowen ratio, the tile approach achieves appropriate results. To illustrate these differences, Fig. 7 shows mean daily cycles for the Waldstein-Weidenbrunnen forest site and the heterogeneous Köhlerloh clearing, whereby every surface contributes 50 % to the additionally shown weighted flux. The magnitudes of the sensible heat flux are, in general, higher at the forest than at the clearing. In contrast, during the day latent heat flux is higher for the forest and
during night the dewfall is higher for the clearing. The main differences for both surfaces occur for the NEE. The magnitude of the NEE for the forest is sometimes more than twice of that from the clearing during the day. During the night, the differences between the NEE for the forest and the clearing are smaller. Falge et al. (2017) already showed a high variation of the NEE between the different species considered in the tile approach for the clearing. This variability is also present for the comparison of the NEE for the forest and the clearing. These differences are illustrated because the NEE data measured at the Waldstein-
Weidenbrunnen site are used in many modeling studies within FLUXNET. For specification of surface characteristics these studies often use remote sensing data, where the forest and the clearing is contained in the relevant grid. Thus, the satellite derived LAI and fractional PAR (Photosynthetically Active Radiation) have to be interpreted as a mixture of forest and clearing. However, the FLUXNET station at Waldstein-Weidenbrunnen is characterized only as a forest measuring station, with *Picea*

*abies* as the main vegetation type, because the fluxes are obtained at the Main Tower (MT, cf. Fig. 1b) or at the Turbulence Tower (TT) since 2007. Consequently, the NEE can hardly be reproduced through consideration of the weighted vegetation parametrization.

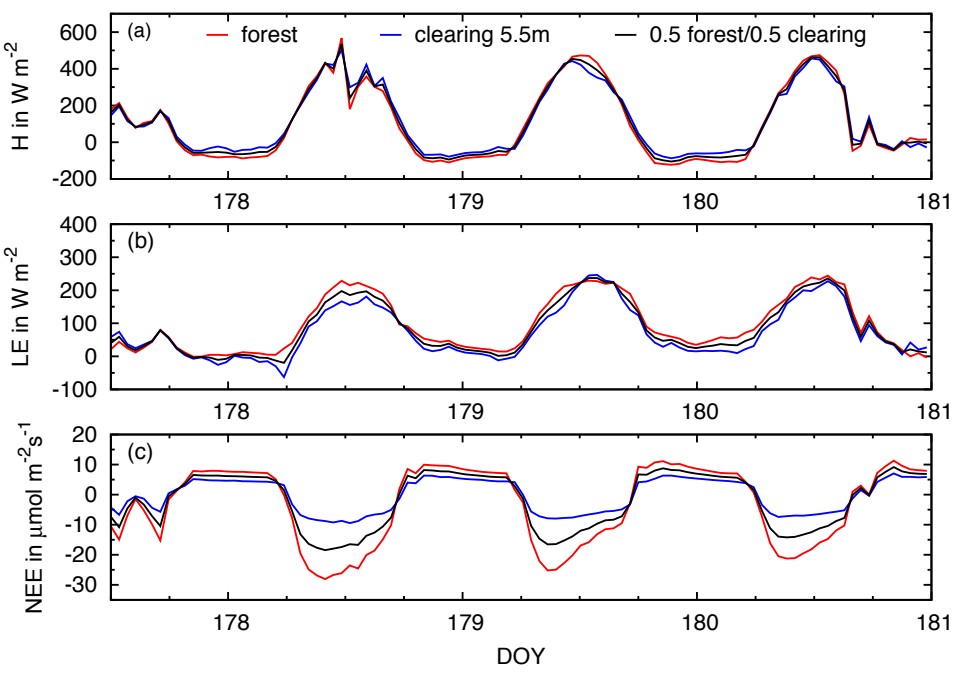

**Figure 7.** Main daily cycle of the a) sensible and b) latent heat flux as well as c) NEE, for the first Golden Day Period modeled with ACASA for the forest (red line) and the clearing 5.5 m (tile approach for the different land use types of the clearing; blue line) and for a flux weighting of 50 % clearing and 50 % forest (black line).

## 4    Conclusions

The ACASA model, which was originally developed for tall vegetation, can also be used with high accuracy for low vegetation, as was demonstrated for the clearing, if the plant specific parameters are appropriately implemented in the model. Therefore, it is applicable to consider ACASA in tile approaches for heteorogeneous areas for tall as well as short vegetation or clearings. The footprint averaged tile approach did not produce significantly better results than the tile approach for the whole clearing, as was done by (Falge et al., 2017). The footprint model is probably not accurate enough in the location of the effect levels of the footprint, considering the small-scale heterogeneities of the clearing in comparison to the size of the footprint area. This issue has already been shown by comparison of different footprint models by Markkanen et al. (2009). For larger heterogeneities, this approach may be more appropriate. Furthermore, the clearing area contained random-like distributed very wet parts, which could not be classified in the land-use characteristics of the tile approaches.

The small underestimation by the model of fluxes above the forest is probably a local overestimation of the fluxes at the Turbulence Tower TT, which is adjacent to an area of the forest where higher fluxes are possible (Foken, 2017b). Large Eddy Simulation studies have shown that at a distance of ten times the canopy height from the forest edge, higher fluxes are possible (Kanani-Sühring and Raasch, 2015). Such a highly local effect is impossible to model in ACASA.

Assuming that the ACASA model is well parametrized and the available energy is accurately distributed to the sensible and the latent heat flux, a good agreement has been found with the energy balance corrected measurements. The correction with the buoyancy flux leads to better results, but this depends on the Bowen ratio, i.e., for $Bo > 1$ it is better than for $Bo < 1$. This result supports earlier findings (Ingwersen et al., 2011; Babel et al., 2014). The idea behind the buoyancy correction, that buoyancy is the reason for secondary circulations, is only partly true. Possible errors of the eddy-covariance method cannot

be inferred from the improved applicability of the Bowen ratio correction for low Bowen ratios ($Bo < 1$). However, a recent investigation by Gao et al. (2017) showed a potential phase shift between the vertical wind component and the scalars, which might bias the determination of turbulent fluxes. This should be an objective of a further study with a data set containing more data for $Bo < 1$. The correction of the NEE is probably useful, but the effect is not very significant for $Bo > 1$, however, it might be more applicable for accumulated fluxes.

*Code and data availability.* An overview of the instrumentation and important measurements at the Waldstein-Weidenbrunnen site is provided by Foken (2017a). Data is available from Thomas Foken on request. The ACASA model code is available from Kyaw Tha Paw U on request.

*Author contributions.* The paper is based on the master's degree thesis of K. Gatzsche, which was advised by A. Raabe and T. Foken and supported by W. Babel (application of the footprint modeling and turbulence data calculation), E. Falge (plant parameters), and D.R. Pyles

and K.T. Paw U (application of ACASA). The additional analysis in this paper and the writing was done by K. Gatzsche advised by T. Foken.

*Competing interests.* The authors declare that they have no conflict of interest.

*Acknowledgements.* The first author acknowledges Prof. Ina Tegen for providing the opportunity to finalize this paper at the Leibniz Institute for Tropospheric Research. We thank the editor and the three anonymous reviewers for the helpful comments.

This research was funded within the DFG projects FO 226/16-1 and ME 2100/4-1 as well the DFG PAK 446 project, mainly the subprojects

ME 2100/5-1 and FO226/21-1, the German Federal Ministry of Education, Science, Research and Technology (PT BEO 51-0339476 D), and BaCaTeC (Bayerisch-Kalifornische Hochschulzentrum) "Modellierung des Energieaustausches zwischen der Atmosphäre und Waldökosystemen". Partial support came from a grant from the US National Science Foundation EF1137306 to the Massachusetts Institute of Technology,

sub-award 5710003122 to the University of California Davis. This publication was funded by the German Research Foundation (DFG) and the University of Bayreuth in the funding programme Open Access Publishing.

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
