# Peer review of "Application of the ACASA model for a spruce forest and a nearby patchy clearing"

_Biogeosciences, 2017_

## Referee Comment (RC1) · Anonymous Referee #1 · 4 Jan 2018

GENERAL COMMENTS:

In this paper, Gatzsche et al. test a third-order closure one-dimensional SVAT model (ACASA) for the prediction of heat, water and $CO_2$ exchanges over a heterogeneous clear-cut and a nearby spruce forest. They compare the modeled fluxes to turbulent flux measurements on a dozen days obtained in the middle of the vegetation season (July). Their specific aims are (i) to test the implementation of a so-called tile approach on the clearcut (weighting the modeled flux contributions of individual land-use according to different schemes to obtain the total modeled flux to be compared to the measured flux) and (ii) to test different scaling of measured turbulent fluxes to correct for underestimation of these fluxes diagnosed through a deficit in the energy balance closure. This scaling is not only applied to heat fluxes but also to $CO_2$ flux, postulating

scalar similarity between heat and $CO_2$ transport. For this second objective, both the forest and the clearcut datasets are used.

This work is a follow-up of recently published research by Falge et al. (2017) on the same site and using the same datasets. More specifically Falge et al. propose in their section 16.3.4 ACASA simulations for the forest and the clearing. They show that ACASA is performing well for predicting heat, water and $CO_2$ exchanges for the forest and the clearing as well, using the "coarser" tile approach for the clearing, i.e. weighting the vegetation specific modeled fluxes by the percentage of land cover for the individual vegetation type within the whole clearing (with the eddy-covariance dataset at 2.25 m). They also test the two scaling proposed in the present paper (EBC-Bo and EBC-HB), already stating that the EBC-HB gives better results than the EBC-Bo.

Therefore I'm wondering what are the real additional and useful finding proposed in the present paper. Regarding objective (i) a finer weighting is proposed, using the respective footprint contribution of each individual vegetation type rather than its percentage of land-cover in the whole clearing. However, this finer approach did not give significantly different results than the "coarser" one because "the footprint model is probably not accurate enough in the location of the effect levels of the footprint, considering the small-scale heterogeneities of the clearing in comparison to the size of the footprint area." (P20L8-9). Looking at the land-cover map provided in Fig 2, this result was very predictable to my opinion. Regarding objective (ii), the present paper investigates more deeply the conditions under which the EBC-Bo and EBC-HB seems to perform well (fig. 6). This goes beyond Falge et al. 2017. These points are explained in section 2.6 and previous papers on which the research is relying are duly cited but I had a hard time disentangling new findings from already published ones. Previous findings and precise objectives should be crystal clear already after reading the introduction which is not the case in the submitted manuscript. This flaw is also reflected in the manuscript title which is vague. Also, the added value of analyzing in the same manuscript the forest and the clearing should be explained in the introduction.

I have also some concern on the quality of the writing. Repetitions, lack of precision in some places, some cumbersome within-paragraph structure. See the numerous specific comments for details. I also found that the manuscript was not enough self-standing, being too much elliptic on important concepts (e.g. the "tile" approach is not defined and the reader is referred to Molders 2012 which is a full book; see also specific comments).

Also, what is the interest of introducing the turbulent flux measurements in the clearing at 2.25 m? They had almost nothing to the story. Cannot it be removed from the manuscript? Same question for the turbulence tower (TT). It is not clear where these measurements have been used (P5L1 does not mention it but in the conclusion, it's mentioned).

Finally, section 3.2.4. is an evidence and I suggest to remove it. The fact that combining a robust model with the tile approach to simulate fluxes having large-scale heterogeneous land-cover within the footprint is quite obvious and does not deserve a fig. and this subsection. In addition, illustration of the different NEE for forest and clearing has already been presented in Falge et al., 2017 (fig. 16.15).

Only after these comments have been taken into account, it will be feasible to estimate whether a "critical mass" is reached to justify a full paper.

SPECIFIC COMMENTS:

P1L19: 30% of unclosure for which situation (mean on a lot of sites?)

P2L12: I'm confused by the use of the term "forest-clearing transition". Do you mean that your fluxes are both (MT and TM) affected by coherent structures because the two towers are close to the forest-clearing transition? After reading your paper, I rather had in mind that the forest tower fluxes were only weakly affected by the presence of the clearcut and that the clearcut fluxes were also only weakly affected by the presence of the forest. So rather than applying the model to a forest-clearing transition, you apply

it for a forest and for a clearing.

P2L11: Group the two paragraphs, you are developing the same idea.

P2L17-18: repetition. Already stated on line 6-16. You can delete this sentence and introduce the refs elsewhere.

P2L19-20: "Additionally, it is evaluated whether the energy balance closure corrected flux measurements better fit the fluxes simulated by ACASA". This objective is embedded in the previous one so which are you using "additionally"?

P2L21: "Field measurements of the FLUXNET site 'Waldstein-Weidenbrunnen' (DE-Bay) were therefore complemented by additional measurements". Which ones? Be more precise.

P2L24: "to model the energy and CO2 exchanges of different vegetation types"

P2L28: "The experimental data for the initialization of the model and the comparison of the results" could be replaced by "The experimental data for the initialization of the model and the evaluation of its outputs".

P3L13-14: "where z is the measurement height normalized by the stand height hc". You probably mean: "where z is the measurement height and hc is the stand height".

P3L13-14: "The understory comprises two-thirds crinkled hairgrass (Deschampsia flexuosa) and moss (together LAI of 0.5m2 m-2 and less) and one third characterized by blueberry (Vaccinium myrtillus) and young Norway spruce (Picea abies, together PAI of 3.5m2 m-2)". Two thirds and one-third on which basis? And what means "and less"?

P4L10: "In the majority of cases, high-frequency gas analyzers for carbon dioxide (cCO2 ) and water vapor (q) were installed in conjunction with sonic anemometers". Why "in the majority of cases"? Please rephrase.

P5L23: "Bell-Berry stomatal conductance". I guess you mean "Ball-Berry"

[Figure]

P5L17-18: "This method allows the calculation of the temperatures of these components without also making substantial errors in the case of significant deviations from the ambient temperature". Hardly understandable.

P6L3-4: R0 is defined as the respiration rate at 0°C but Equ. 1 will not give RT(Ts=273K)=R0. Probably Ts should be expressed in degree Celsius instead of Kelvin.

P7L22: "For the correction of the energy fluxes, the residual (Res) arises from the following assumption:". Equ. 3 that follows is the definition of the residuals, I don't see any assumption there.

P7L25-27: not necessary to cite Haverd and Lindroth twice. Please reorganize.

P7L29: "with mf the biomass of the forest". You mean the above-ground biomass?

P9L1-2: "This method is usually utilized for the correction of heat fluxes under the assumption of measuring errors, ...". Repetition from the previous sentence. Please rephrase.

P9L9: "The discrepancy between measured and simulated NEE can be an effect of the unclosed energy balance on the $CO_2$ fluxes". I understand what you mean but this is a complicated way of saying that if $CO_2$ exchanges share the same transport processes than heat exchanges (scalar similarity), measured $CO_2$ exchanges should be underestimated on the same level as heat fluxes. And this hypothesis being far from widely accepted, this point should be discussed further.

P9L20: "whereby a spectral method of the flux averaging of surface characteristics (roughness length) according to Hasager and Jensen (1999) is employed". I do not understand this part of the sentence. Please be more explicit.

P10L5-6: "However, it has been found that the energy balance closure for the sensible heat, the latent heat, and the NEE was better for the buoyancy flux correction, but the results are partly inconclusive". Be more precise, what mean "partly inconclusive"?

See my general comment on this point.

P10L15-19: This §should be reorganized to avoid repetitions.

P10L23: Remove "thus", there is no causality link with the previous sentence. You just start the explanation of the previous sentence.

P10L26-28: You probably switched westerly and easterly!

P10L29-30: Why didn't you use only the common dataset at the two heights? It would avoid this problem.

P11L4-5: Use or at least recall the acronyms defined in section 2.4 (EBC-Bo, EBC-HB), it will be more explicit than "the sensible heat flux corrected with the buoyancy flux".

P12L2-3: "Obviously, the integrated fluxes of the tile approaches for the whole clearing and the footprint of the turbulence mast for 5.5m height do not differ significantly.". I understand what you mean but literally you compare apples and pears. Please rephrase.

P14L24-26: "Due to Bo > 1, the buoyancy correction overestimates the effect of thermal convection on the energy balance closure and the true correction might lie between both correction methods". I do not understand this sentence. Why EBC-HB should overestimate H when H is dominant? Is the explanation linked to the following sentence?

Fig. 5: Would be convenient if all the y-axis scales would be identical for the forest and the clearing. Would also be convenient if the colors/ markers used would be compliant with those udes in fig. 3 and 4.

P17L9: "This could also be an overestimation by the measured fluxes due to the turbulence and the forest structure, discussed by Foken (2017b)". Please explain what you mean, in order to have this manuscript self-standing. It's a bit more explicit in the conclusion but should be moved here.
P18L2: "According to the findings in Sect. 3.2.2 that a large contribution of the un-closed energy balance is a missing sensible heat flux, we used the modeled data as a reference for the validation of the correction methods". I guess that the first part of the sentence should be deduced by the reader based on fig. 5. It would be better to comment this feature already in sect. 3.2.2. Also, I do not understand the link between the first part of the sentence and the second part. Whether is measured H or measured LE the main responsible for the non-closure of the energy balance, you can decide to use the modeled values as the reference. Please clarify.

P18L15-15: "Additionally, this indicates that the Bowen-ratio correction is not a method that is applicable for the correction of the measurement errors that occur". Confusing since the reader does not know if this assertion holds for a given range of Bo or for the whole range.

P18L17-18: "Due to the assumption of a similarity between the water and carbon diox-ide fluxes (Ruppert et al., 2006), neither the NEE flux nor the latent heat flux were corrected for high Bowen ratios". Not necessary to cite again Ruppert et al. at this stage. Also, this sentence is not well integrated in the discussion and is ambiguous. I would prefer something like: "In all cases, attribution of residual energy to latent heat flux is low for high bowen ratios. Therefore, due to the assumption of a similarity be-tween the water and carbon dioxide fluxes, the NEE flux was only marginally corrected in these conditions".

P19L4-5: "Due to high Bowen ratios and large underestimation by the model, the buoy-ancy corrected fluxes show better results in comparison with the model". This sentence brings nothing in this discussion, I would simply delete it. If you think it brings neces-sary information, please improve it to be more explicit.

P20L21-23: "The better correction . . . could be a reason". Not understandable for the reader. Please rephrase/improve. And consider revisiting the writing style of the whole conclusion to avoid this "telegraphic style".

TECHNICAL CORRECTIONS:

P1L16: replace "stratification" by "atmospheric stratification".

P10L15: "of the" appears twice.

Fig 2: Distance units are missing.

Fig. 4: you can simplify the legend by "same as fig 3 but for the second GDP".

[Figure]

---

## Referee Comment (RC2) · Anonymous Referee #2 · 5 Jan 2018

Review to the manuscript "Application of the ACASA model for a spruce forest and a nearby patchy clearing" by K. Gatzsche et al.

General comments: First, I want to thank the referee No. 1 for the very detailed review. In the following, I do not repeat all these helpful comments. I am going to concentrate my issues on the application of the model ACASA. As already mentioned by the first referee, the manuscript by Gatzsche et al. is a continuation of former works. I was already involved in the review process of former papers. Thus, I have also doubts concerning the distinct novelty of the presented manuscript. Furthermore, I have serious concern about the applicability of the one-dimensional ACASA model for the heterogeneous site of a clearing. The model core of ACASA is based on a 1D boundary layer model with third-order turbulence closure according to the 30 years old papers of

[Figure]

Meyers and Paw U. The authors must explain 1. how this kind of model can be used for the highly heterogeneous flow and turbulence field (due to lateral exchange and influences!) 2. which assumptions have to be applied to consider the heterogeneity of the experimental site, and 3. which special conditions must be held to minimize the influence of heterogeneity on fluxes (e.g., extension of the clearing)

In later studies, different extensions of ACASA concerning the "biological" part of the model were implemented, but the model core remained unchanged during this time. 1D atmospheric models are limited for applications on extended homogeneous landscapes. The inclusion of high-order turbulence closure into the atmospheric model permits a more realistic consideration of the vertical turbulent exchange by inclusion of higher-order moments in relation to models with K-closure. But lateral exchange effects like the advection due to edge influences (e.g., due to the transition zone between clearing and forest) cannot be calculated directly by 1D models and must be parameterized. This leads, among other problems, to a strong dependence of the simulation result on wind direction and thermal stratification. The tile approach used in this manuscript presumes that the advection effect on the weighted average of fluxes is negligible. A good match between the ACASA simulations and the measurements can be reached by model "tuning" regardless of reasons for discrepancies - but in this case the transferability of the model results is more than uncertain. As the state of the art, 3D models (LES, RANS, LES-RANS-Hybrids) are used to tackle the problem of advection. The authors must discuss this lack of ACASA more detailed including comparative papers for 3D models (e.g. Sogachev et al., Tellus 54B, 2002 or Sogachev and Lloyd, BLM 112, 2004 or Hanjalic, J. Fluids Eng. 127(5):831-839, 2005) and they must give an estimation of the effect of advection (induced by the heterogeneity of the measurement site) on the uncertainty of model simulation.

Specific comments: p.4: "Due to the heterogeneous vegetation at the clearing, no distinct mean canopy height can be estimated for the clearing" Which canopy height was used as input in ACASA for the clearing in this case? p.6: eq. 1: The original

parameterization comes from another climate region in North Carolina. Is there an adaptation of parameterization before using the equation in ACASA? p. 16/17, tables 4 and 5: Typically, $r^2$ and degree of significance must be given by the authors.

---

## Editor Comment (EC1) · A. Ibrom (Editor) · 6 Jan 2018

Dear Authors

you have received two careful reviews from two out of three possible referees. Both raise concerns about the novelty of your work and give clear explanations why. I'm convinced that you deem your work being a novel development from previous work, but it looks as if this was not yet expressed in an optimal manner in your manuscript. I'd like to encourage you to use the interactive discussion to sort out with the referees, where the presentation didn't work out well. You are welcome to use the interactive discussion to comment and ask on particular, important aspects in order to receive immediate, constructive responses and clarifications, where necessary.

With kind regards,

Andreas Ibrom

---

## Referee Comment (RC3) · Anonymous Referee #3 · 8 Jan 2018

The paper investigates the possibilities to apply the ACASA Model for the calculation of turbulent fluxes over heterogeneous landscape consisting of forest and clearing. The paper clearly presents the results of a model development study. That's why the title of the article is slightly misleading in my option – the "application of the model" usually means that we use a model as an instrument to study something, e.g. turbulent fluxes. In such case the model is a tool not the goal. In present article the "overall aim" is to "to analyze whether ACASA can simulate the fluxes over tall and low vegetation in an appropriate way", i.e. the model itself is the aim. Therefore, I think it would be better to resubmit the article to some journal focused on model development like "Ecological modelling".

However, whether the publication will take place in this or in a modelling journal, it

should be clearly defined what is the main aim of the proposed model development. What do you plan to use the model for? If you want to study the processes in a spatially heterogeneous landscape than it would be better to use a 3D model as an instrument. We cannot learn much about the 3D effects in the spatially heterogeneous landscape using the 1D model. The authors write that the model "has been utilized for the simulation of the turbulent transfer for a forest-clearing transition." It is not true in my opinion - what was described are the forest AND the clearing, but not the transition. The authors stated themselves that for such studies a 3D approach like LES modelling is better. Or is the aim of the model to use it as an enhanced land surface submodel for climate modelling?

Other questions are: what are the main improvements of the corrected model and whether they are sufficient for publication? The authors state that the present paper is "an updated investigation" of the Falge et al (2017), so what is their new and original contribution? For the model improvement several already published methods were implemented and the share of "we suggested" seems to be relatively small. Please highlight clearly the new and original achievements of the study.

Besides, the text should be thoroughly checked - there are many mixings between singular and plural, and the formulations in the text are not really clear (s. also the special remarks below).

Special remarks:

P1 L14 "models must be applied for different surface types" Comment: It is not quite clear why it is a MUST to apply any model at all and for different surfaces in particular?

P1 L19 "not closed by an amount of up to 30%" Comment: 30% of unclosure is not an absolute maximum, is it?

P2 L2 "Measurement data must be corrected..." Comment: again – is it always a MUST?

**BGD**

P2 L25 "the footprint of different plant classes." Comment: What do you mean?

P3 L9 "more detailed description of the surrounding topography is provided by Foken…" Comment: It is an important information for a study dealing with heterogeneous surface and should be provided in a manuscript.

P3 L12 "The main foliage is located" Comment: What is the main foliage? Do you mean a leaf area or mass or what?

P3 L12 "where z is the measurement height normalized by the stand height hc" Comment: z is just a measurement height.

P4 Table 1. Why clearing is described in the table 1 and forest – in the text?

P3 L12 "This method allows the calculation of the temperatures of these components without also making substantial errors in the case of significant deviations from the ambient temperature" Comment: 1) Why "also"? 2) Does it work for the components in a forest-clearing transition zone?

P5 L19 "Direct as well as diffuse radiation can be absorbed, transmitted, emitted or reflected by the canopy, whereby these processes are dependent on the leaf distribution of the plants." Comments: 1) The terms "direct" and "diffuse" are usually applied for shortwave solar (or diffuse sky) radiation fluxes which, therefore, cannot be emitted by the canopy under normal conditions. The canopy can emit a longwave radiation. 2) Why only the leaf distribution? What about stems, branches and soil?

P5 L20 "the vegetation is distributed in 10 different leaf angle classes" Comments: the whole vegetation or just leaves/needles?

P5 L21 "allocation of the incoming energy" Comments: or of the available? The emitted/reflected parts of incoming energy are lost for the canopy.

P5 L25 How tall is "taller vegetation"?

P5 L26 What are the "diffusive approaches"?

[Figure]

P6. Table 2: The measured parameters and instrumentation are not presented adequately: - What is the instrument for the air pressure? - CM14 is an albedometer and can measure both incoming and reflected shortwave radiation. Did you use the incoming component only? - CG2 – is the net radiometer, thus, the same question as for the CM14 - CNRN4 is probably the CNR4 net radiometer – the same questions about upward and downward fluxes. If you measured the downward fluxes only – how did you calculate the radiative balance? What is the instrument for the soil heat flux? Is there any instrument for PAR-measurements?

P6 L4-5 "The soil respiration is separately calculated for 5 roots and microbes according to Eq. (1)" Comment: using the same R0 and Q10?

P7 L7 "the relation" – which relation?

P7 L13-14: As the information is important, I suggest to provide both parameter sets – clearing and forest for the better comparison and visualization of differences.

P7 L18 – PAR, measured or modelled? P7 L27 – Second reference of Haverd and Lindroth is not necessary P9 L3-4 "...but requires their scalar similarity. Henceforth, this method is abbreviated with EBC-Bo." Comment: The formulation is unclear – why scalar similarity produces this abbreviation? The same question is about EBC-HB.

P9 L10 "Therefore, we propose a new correction method on the basis of the good scalar similarity between the humidity and the carbon dioxide concentration (Ruppert et al., 2006). Comment: Again, the formulation is unclear – is it your original suggestion or the one of the Ruppert et al?

P9 L20 "of surface characteristics (roughness length)" – characteristic or characteristics?

P10 L22 "between the different stratification" – singular or plural?

P10 L24 " forest has additional influence" – how is it visible in Fig.2?

P11 Fig. 2: Consider placing the four (a) panels (upper row) below the four (b) panels (lower row) for a direct comparison of footprints. What is the "all" case?

P11 L7-8. The sentence is not clearly formulated please rewrite. What is the "land-use distribution of the whole clearing"? Do you mean different land covers or vegetation types?

P11 L11-13"…difference… is more dominant" – please reformulate.

P12L2-3 "Obviously, the integrated fluxes of the tile approaches for the whole clearing and the footprint of the turbulence mast for 5.5m height do not differ significantly" – what do you compare with what?

P12 L3: Which "land use characteristics"?

P12 L4-5 "This can be seen by the high measured latent heat fluxes in the first Golden Day 5 Period (see Fig. 3). Further regression analyses…" Comment: Why "further" – there is no regression analysis on Fig.3?

P12L3 Fig. 3. What about uncorrected fluxes?

P13 L3 "regression has been achieved" – can you achieve a regression?

P13 L3 "regression analysis is calculated for" – can you calculate an analysis?

P14 L1-2 "difference between net radiation and the ground heat flux" – how they were measured?

P14 L18 – "sensors" – which sensors?

P18 "Further research is necessary with other well parametrized models that close the energy balance very well, and data sets for lower Bowen ratios." Comment: does it mean that the present model is not adequate for the goal?

P19 L2 "The comparison of the modeled and the measured (ground truth)" Comment: in previous section the ground truth was the model, wasn't it? P19 L5-6 "the buoyancy

corrected fluxes show better results in comparison with the model. This offers the possibility of modeling fluxes over larger heterogeneous areas. . ." Comment: This does not look like a logical consequence for me.

P20 L4 "can also be used with high accuracy for low vegetation if the plant specific parameters are appropriately implemented in the model" Comment: What in this case is the "high accuracy" and "low vegetation"? The conclusion does not follow from the results in my opinion.

P20 L7-12. The paragraph is not really clear. Try to reformulate it.

---

## Author Comment (AC1) · 24 Jan 2018

Dear Dr. Ibrom,

Thank you very much for your comment and thanks to the reviewers for their comments. In response to the opinions that the final results of the reviewers are different from the main revisions (Rev. #1), the model is not up to date (Rev. #2), Biogeosciences is the wrong journal (Rev. #3) and the paper is similar to Falge et al. (2017), we are providing you and the three reviewers with some answers to the comments. Most of the specific points can be easily corrected or explained without significant consequences for the manuscript.

1. We agree that the paper looks quite similar to the publication by Falge. The Falge paper is a summary of the results of all model studies made in the last nearly 20 years at the Waldstein-Weidenbrunnen site, without any discussion of specific problems. However, reviewer #1 has selected the two points that are new in the present manuscript: (i) a more precise concept of the comparison of flux measurements and their footprints and 1D modeling with a tile approach; (ii) a detailed discussion of the effects of different energy balance correction schemata. We agree that we should highlight both points more concretely in the first two paragraphs of the introduction (P1L13 – P2L5 of the submitted manuscript).

2. As was mainly stated by **reviewer #2,** the 1D model ACASA is very old and not appropriate to our measuring site. **Here we disagree**. It is right that the basics of this model – mainly the higher order closure concept – were developed more than 30 years ago (Meyers and Paw U, 1986, 1987). The model was significantly updated by the University of Davis (Pyles et al., 2000; Pyles et al., 2003) and it is still in use (Falk et al., 2014). About ten years ago we started to work with ACASA. The first issue was to use a sensitivity analysis to check whether the model can be applied for a Central European spruce forest (Staudt et al., 2010). We also found some model-specific problems (Staudt, 2010), and the model was again updated by the University of Davis. Using the updated model, a first study for the Waldstein-Weidenbrunnen site was published (Staudt et al., 2011). This study showed that the ACASA model is more accurate (in comparison to a K-approach model) when coherent structures dominate at night. We have not highlighted coherent structures (see reviewer #1, P2L12), because a paper providing an overview of about 15 years research on this topic is available. We will include the reference (Thomas et al., 2017). The model was changed by Staudt et al. (2011) for the spruce-specific parameterizations, and Falge et al. (2017) determined several parameters of the plants of the clearing (Table 16.1). This history of the ACASA model is partly described in the first part of Section 2.3 of our paper. In the second part we have listed all changes made in the model according to Staudt et al. (2011) and Falge et al. (2017) for the clearing. We believe that this second part of Section 2.3 is necessary to ensure that the reader does not assume that we only used a version of the model developed by the University of Davis without site-specific parameterizations. From the given references in Section 2.3 it should be clear that we used a well-developed model and our paper is not a modeling study. In this point we **disagree with the position of reviewer #3** that the paper is not appropriate for Biogeosciences. We copied Table 16.1 from Falge et al. (2017) in our paper (Table 3) because we believe that these data are relevant for the understanding of the tile approach and that the reader should not need a book chapter for a complete understanding of our results; however, Table 3 may not be necessary.

3. According to the comments of all three reviewers, we found that we have not explained our measurement and model comparison concept precisely enough, which had only been done

with some references, e.g. the definition of the tile approach (e.g. reviewer #1): The fluxes measured with the eddy covariance technique are related to all surfaces on the up-wind side of the measurements, and the influence of each surface on the measured data is given by the footprint function. For further details see Leclerc and Foken (2014). If you want to compare these flux measurements with 1D models, this can be done over a homogeneous surface easily or over a heterogeneous surface if the model is parameterized for all surface characteristics and averaged according to the percentage of each surface in the footprint (tile approach). The footprint is due to changing wind velocity and atmospheric stratification that are different for each measurement (about 30 minutes), and the tile approach is also different for each measurement. The deficit of the Falge et al. (2017) paper was that all measurements were compared with the same tile approach for the clearing. This problem was solved in the presented paper. This concept works quite well for only two significantly different surfaces in the footprint (Foken and Leclerc, 2004; Göckede et al., 2005; Biermann et al., 2014), but in our case, with multiple surface characteristics, it partly fails. We will discuss this issue in more detail (now already P20L7-12) in comparison with the sensitivity study for footprint models (Markkanen et al., 2009).

4.  The term "forest-clearing interaction" (P2L12) probably confused all reviewers and they assumed an advection study. We will delete this term, because the tile approach is not an advection study (for forest clearing interaction see e.g. Foken et al. (2017)). Reviewer #1 has indicated that the paper is not a "forest-clearing interaction" and proposed the deletion of Section 3.2.4. We would not delete it, but we have to give some more explanations of why it should be used and provide Fig. 7, which is only for modeled NEE similar to Fig. 16.15 of Falge et al. (2017). The NEE data measured at Waldstein-Weidenbrunnen site are used in many modeling studies within FLUXNET. For surface characteristics these often use MODIS data with a grid size of 500 m or 250 m. Since 2007, the satellite sees about 50 % forest and 50 % clearing in the relevant grid, and the authors compare these data with measured fluxes that have nearly 100 % forest in the footprint. We will highlight the problem and reformulate Section 3.2.4. Reviewer #2 and #3 propose the use of a 3D model rather than a 1D model. However, this would also need a different measurement strategy to directly measure advection. Up to now, experimental advection studies with an acceptable number of measurement points failed (Aubinet et al., 2010; Aubinet et al., 2012). Furthermore, the resolution of available 3D models (Sogachev et al., 2002; Sogachev and Lloyd, 2004) is too large (25–50 m) for the small scale heterogeneities in the clearing (5–10 m). By the way, a 3D LES study (model PALM) with 1 m grid resolution was made for the site, but only for a short time series (Kanani-Sühring et al., 2017); see also P20L14–16. Because our study is a comparison of experimental data and modeled data, both concepts must agree (see point 3). Furthermore, ACASA and PALM are available for free while most of the 3D models are commercial models. On the basis of these points **we disagree with reviewers #2 and #3** that the ACASA and footprint modeling should be replaced by 3D modeling. Footprint and 1D models (SVAT) are still widely used.

5.  Besides the comparison of the footprint – tile approach for the clearing, the energy balance closure study was done in much more detail than was the case by Falge et al. (2017). In particular, the conclusion made by Falge et al. (2017) for both correction methods was strengthened. New is the proposed correction for NEE. Therefore, the scalar analysis by Ruppert et al. (2006) is necessary. We are sorry, but the title of this paper is confusing. By the

way, Foken (2008) is an overview paper about the energy balance closure problem and not a special field study.

We hope we have been able to explain why we disagree with the conclusions of reviewers # 2 and #3. If you agree, we would like to follow the proposed revision of reviewer #1 and apply the specific comments by the other reviewers. In this case, we would like to submit a revised paper within one month.

Sincerely,

Kathrin Gatzsche and Thomas Foken for all authors

References:

Aubinet, M., Feigenwinter, C., Heinesch, B., Bernhofer, C., Canepa, E., Lindroth, A., Montagnani, L., Rebmann, C., Sedlak, P., and van Gorsel, E.: Direct advection measurements do not help to solve the night-time $CO_2$ closure problem: Evidence from three different forests, Agrical. Forest Meteorol., 150, 655-664, 2010.

Aubinet, M., Feigenwinter, C., Heinesch, B., Laffineur, Q., Papale, D., Reichstein, M., Rinne, J., and Van Gorsel, E.: Nighttime flux correction, in: Eddy Covariance: A Practical Guide to Measurement and Data Analysis, edited by: Aubinet, M., Vesala, T., and Papale, D., Springer, Berlin, Heidelberg, 133-157, 2012.

Biermann, T., Babel, W., Ma, W., Chen, X., Thiem, E., Ma, Y., and Foken, T.: Turbulent flux observations and modelling over a shallow lake and a wet grassland in the Nam Co basin, Tibetan Plateau, Theor. Appl. Climat., 116, 301-316, 2014.

Falge, E., Köck, K., Gatzsche, K., Voß, L., Schäfer, A., Berger, M., Dlugi, R., Raabe, A., Pyles, R. D., Paw U, K. T., and Foken, T.: Modelling of energy and matter exchange, in: Energy and Matter Fluxes of a Spruce Forest Ecosystem, Ecological Studies, Vol. 229, edited by: Foken, T., Springer, Cham, 379-414, 2017.

Falk, M., Pyles, R. D., Ustin, S. L., Paw U, K. T., Xu, L., Whiting, M. L., Sanden, B. L., and Brown, P. H.: Evaluated crop evapotranspiration over a region of irrigated orchards with the improved ACASA-WRF model, J. Hydrometeorol., 15, 744-758, 2014.

Foken, T., and Leclerc, M. Y.: Methods and limitations in validation of footprint models, Agrical. Forest Meteorol., 127, 223-234, 2004.

Foken, T.: The energy balance closure problem – An overview, Ecolog. Appl., 18, 1351-1367, 10.1890/06-0922.1, 2008.

Foken, T., Serafimovich, A., Eder, F., Hübner, J., Gao, Z., and Liu, H.: Interaction forest–clearing, in: Energy and Matter Fluxes of a Spruce Forest Ecosystem, Ecological Studies Vol. 229, edited by: Foken, T., Springer, Cham, 309-329, 2017.

Göckede, M., Markkanen, T., Mauder, M., Arnold, K., Leps, J. P., and Foken, T.: Validation of footprint models using natural tracer measurements from a field experiment, Agrical. Forest Meteorol., 135, 314-325, 2005.

Kanani-Sühring, F., Falge, E., Voß, L., and Raasch, S.: Complexity of flow structures and turbulent transport in heterogeneously forested landscapes: LES study of the Waldstein site, in: Energy and Matter Fluxes of a Spruce Forest Ecosystem, Ecological Studies, Vol. 229, edited by: Foken, T., Springer, Cham, 415-436, 2017.

Leclerc, M. Y., and Foken, T.: Footprints in Micrometeorology and Ecology, Springer, Heidelberg, New York, Dordrecht, London, XIX, 239 pp., 2014.

Markkanen, T., Steinfeld, G., Kljun, N., Raasch, S., and Foken, T.: Comparison of conventional Lagrangian stochastic footprint models against LES driven footprint estimates, Atmos. Chem. Phys., 9, 5575-5586, 10.5194/acp-9-5575-2009, 2009.

Meyers, T. P., and Paw U, K. T.: Testing a higher-order closure model for modelling airflow within and above plant canopies, Boundary-Layer Meteorol., 37, 297-311, 1986.

Meyers, T. P., and Paw U, K. T.: Modelling the plant canopy microenvironment with higher-order closure principles, Agrical. Forest Meteorol., 41, 143-163, 1987.

Pyles, R. D., Weare, B. C., and Paw U, K. T.: The UCD Advanced Canopy-Atmosphere-Soil Algorithm: comparisons with observations from different climate and vegetation regimes, Quart. J. Roy. Meteorol. Soc., 126, 2951-2980, 2000.

Pyles, R. D., Weare, B. C., PawU, K. T., and Gustafson, W.: Coupling between the University of California, Davis, Advanced Canopy Atmosphere Soil Algorithm (ACASA) and MM5: Preliminary Results for July 1998 for Western North America, J. Appl. Meteorol., 423, 557 - 569, 2003.

Ruppert, J., Thomas, C., and Foken, T.: Scalar similarity for relaxed eddy accumulation methods, Boundary-Layer Meteorol., 120, 39-63, 2006.

Sogachev, A., Menzhulin, G., Heimann, M., and Lloyd, J.: A simple three dimensional canopy-planetray boundary layer simulation model for scalar concentrations and fluxes, Tellus, 54B, 784-819, 2002.

Sogachev, A., and Lloyd, J.: Using a one-and-a-half order closure model of atmospheric boundary layer for surface flux footprint estimation, Boundary-Layer Meteorol., 112, 467-502, 2004.

Staudt, K.: Modeling the exchange of energy and matter within and above a spruce forest with the higher order closure model ACASA, PhD Thesis, Univerity of Bayreuth, Bayreuth, 2010.

Staudt, K., Falge, E., Pyles, R. D., Paw U, K. T., and Foken, T.: Sensitivity and predictive uncertainty of the ACASA model at a spruce forest site, Biogeosci., 7, 3685-3705, 10.5194/bg-7-3685-2010, 2010.

Staudt, K., Serafimovich, A., Siebicke, L., Pyles, R. D., and Falge, E.: Vertical structure of evapotranspiration at a forest site (a case study), Agrical. Forest Meteorol., 151, 709-729, 10.1016/j.agrformet.2010.10.009, 2011.

Thomas, C. K., Serafimovich, A., Siebicke, L., Gerken, T., and Foken, T.: Coherent structures and flux coupling, in: Energy and Matter Fluxes of a Spruce Forest Ecosystem, Ecological Studies, Vol. 229, edited by: Foken, T., Springer, Cham, 113-135, 2017.

---

## Referee Comment (RC4) · Anonymous Referee #2 · 25 Jan 2018

Dear authors, thank you for the detailed reply on the review. I can follow the explanations concerning the actual state and widespread usage of ACASA in the scientific community. It is also visible for me that the authors are willing to deal with the limits of ACASA in simulation of local advection. I would be ready to accept the paper under the following conditions:

The authors must extend the conclusions by a paragraph which describes the limits of ACASA concerning simulation of local advection. For it, the authors should use their answers from the interactive discussion including a sentence about uncertainties of ACASA when 3D effects are neglected. The answer in the discussion that the methods of measurements and simulations must be coincident seems to be plausible for me. But it does not replace an estimation how the local advection contributes to the mea-

sured energy budget in the footprint area. A short comment in the conclusion to that fact is also recommended. Finally the authors must give a comment about the transferability of the newly parameterised ACASA to other sites with a similar heterogeneity of landuse.

---

## Editor Comment (EC2) · A. Ibrom (Editor) · 25 Jan 2018

Dear referees and authors

I have asked for re-opening the interactive discussion for another 14 days in order avoid a slow and tiring iteration process.

Referees, please comment on the recent authors comment but only on the most essential reservations, that you still might have. The more focused the better!

And Authors, please use this opportunity to directly discuss with the referees and locate the main critical remaining aspects. The more focused the better!

Let's hope that this experiment works!

[Figure]

Thank you very much for your cooperation.

Best wishes, Andreas
* * *

---

## Author Comment (AC2) · 25 Jan 2018

Dear referee,

thank you for your comments. We will accept your proposals for the modification of the manuscript.

Sincerely, on hehalf of all authors

Kathrin Gatzsche

---

## Author Comment (AC3) · 28 Feb 2018

First of all we want to thank the three reviewers for the helpful comments. We have already answered the main problems of the reviewers in a letter to the editor on January 24, which was distributed to all reviewers: https://www.biogeosciences-discuss.net/bg-2017-450/bg-2017-450-AC1-supplement.pdf. Therefore, in the following we will answer only the specific problems. In accordance with these general comments we have made large additions to the Introduction and deleted parts about the ACASA-model that were repetitions from earlier publications. Comments from referees are presented in black and these are followed by the authors' responses in blue.

**Reviewer #1:**

**Specific comments:**

P1L19: 30% of unclosure for which situation (mean on a lot of sites?)

P1L19: The paper is an overview paper.

P2L12: I'm confused by the use of the term "forest-clearing transition". Do you mean that your fluxes are both (MT and TM) affected by coherent structures because the two towers are close to the forest-clearing transition? After reading your paper, I rather had in mind that the forest tower fluxes were only weakly affected by the presence of the clearcut and that the clearcut fluxes were also only weakly affected by the presence of the forest. So rather than applying the model to a forest-clearing transition, you apply it for a forest and for a clearing.

P2L12: See general remarks.

P2L11: Group the two paragraphs, you are developing the same idea

P2L17-18: repetition. Already stated on line 6-16. You can delete this sentence and introduce the refs elsewhere.

P2L11, and L17-18: was done.

P2L19-20: "Additionally, it is evaluated whether the energy balance closure corrected flux measurements better fit the fluxes simulated by ACASA". This objective is embedded in the previous one so which are you using "additionally"?

P2L21: "Field measurements of the FLUXNET site 'Waldstein-Weidenbrunnen' (DE-Bay) were therefore complemented by additional measurements". Which ones? Be more precise.

P2L24: "to model the energy and CO2 exchanges of different vegetation types"

P2L19-20, L21, L24: Text was re-formulated and supplemented with additional text.

P2L28: "The experimental data for the initialization of the model and the comparison of the results" could be replaced by "The experimental data for the initialization of the model and the evaluation of its outputs"

P2L28: was done.

P3L13-14: "where z is the measurement height normalized by the stand height hc". You probably mean: "where z is the measurement height and hc is the stand height"

P3L13-14: was corrected.

P3L13-14: "The understory comprises two-thirds crinkled hairgrass (Deschampsia flexuosa) and moss (together LAI of 0.5m2 m-2 and less) and one third characterized by blueberry (Vaccinium myrtillus) and young Norway spruce (Picea abies, together PAI of 3.5m2 m-2)". Two thirds and one-third on which basis? And what means "and less"?

P3L13-14: was re-formulated.

P4L10: "In the majority of cases, high-frequency gas analyzers for carbon dioxide (cCO2 ) and water vapor (q) were installed in conjunction with sonic anemometers". Why "in the majority of cases"? Please rephrase

P4L10: was re-formulated.

P5L17-18: "This method allows the calculation of the temperatures of these components without also making substantial errors in the case of significant deviations from the ambient temperature". Hardly understandable.

P5L17-18: was re-formulated.

P5L23: "Bell-Berry stomatal conductance". I guess you mean "Ball-Berry"

P5L23: was corrected.

P6L3-4: R0 is defined as the respiration rate at 0 °C but Equ.1 will not give RT(Ts=273K)=R0. Probably Ts should be expressed in degree Celsius instead of Kelvin.

P6L3-4: This part was deleted because this was already published.

P7L22: "For the correction of the energy fluxes, the residual (Res) arises from the following assumption:". Equ. 3 that follows is the definition of the residuals, I don't see any assumption there.

P7L22: was re-formulated.

P7L25-27: not necessary to cite Haverd and Lindroth twice. Please reorganize

P7L25-27: was deleted.

P7L29: "with mf the biomass of the forest". You mean the above-ground biomass?

P7L29: was corrected.

P9L1-2: "This method is usually utilized for the correction of heat fluxes under the assumption of measuring errors, ...". Repetition from the previous sentence. Please rephrase.

P9L1-2: We do not see any repetition.

P9L9: "The discrepancy between measured and simulated NEE can be an effect of the unclosed energy balance on the CO2 fluxes". I understand what you mean but this is a complicated way of saying that if CO2 exchanges share the same transport processes than heat exchanges (scalar similarity), measured CO2 exchanges should be underestimated on the same level as heat fluxes. And this hypothesis being far from widely accepted, this point should be discussed further.

P9L9: We do not say that scalar similarity is fulfilled. We say that the methods can only be applied if the scalar similarity is fulfilled. Ruppert et al. (2006) have shown that this is not always the case. Therefore the agreement of measured (corrected) and modelled fluxes is not always fulfilled. We have added a short statement about the reference (Ruppert et al. 2006).

P9L20: "whereby a spectral method of the flux averaging of surface characteristics (roughness length) according to Hasager and Jensen (1999) is employed". I do not understand this part of the sentence. Please be more explicit.

P9L20: We have added some more details: Because of the non-linearity, you have to average the friction and not the roughness length.

P10L5-6: "However, it has been found that the energy balance closure for the sensible heat, the latent heat, and the NEE was better for the buoyancy flux correction, but the results are partly inconclusive". Be more precise, what mean "partly inconclusive"?

P10L5-6: This part was deleted.

P10L15-19: This § should be reorganized to avoid repetitions.

P10L15-19: P10L1-13 were deleted.

P10L23: Remove "thus", there is no causality link with the previous sentence. You just start the explanation of the previous sentence.

P10L23: was deleted.

P10L26-28: You probably switched westerly and easterly!

P10L26-28: was corrected.

P10L29-30: Why didn't you use only the common dataset at the two heights? It would avoid this problem.

P10L29-30: We have deleted the sentence because we cannot see a significant difference between the effect of the different heights and the slightly different numbers of data points.

P11L4-5: Use or at least recall the acronyms defined in section 2.4 (EBC-Bo, EBC-HB), it will be more explicit than "the sensible heat flux corrected with the buoyancy flux".

P11L4-5: We have added the acronyms.

P12L2-3: "Obviously, the integrated fluxes of the tile approaches for the whole clearing and the footprint of the turbulence mast for 5.5m height do not differ significantly.". I understand what you mean but literally you compare apples and pears. Please rephrase.

P12L2-3: was corrected.

P14L24-26: "Due to Bo > 1, the buoyancy correction overestimates the effect of thermal convection on the energy balance closure and the true correction might lie between both correction methods". I do not understand this sentence. Why EBC-HB should overestimate H when H is dominant? Is the explanation linked to the following sentence?

P14L24-26: The sentence is linked to the sentence before and therefore to the latent heat flux.

Fig. 5: Would be convenient if all the y-axis scales would be identical for the forest and the clearing. Would also be convenient if the colors/ markers used would be compliant with those udes in fig. 3 and 4.

Fig. 5: The scaling has been corrected. The markers/colors of Fig. 5 cannot be directly compared to Fig. 3/4, which includes two weighting methods plus the corrected fluxes (4 markers), where Fig. 5 compares the footprint-weighted tile-approach with the uncorrected, the buoyancy corrected, and the Bowen-ratio corrected fluxes. The colors used in Fig. 5 are compliant with the colors utilized in Fig. 6.

P17L9: "This could also be an overestimation by the measured fluxes due to the turbulence and the forest structure, discussed by Foken (2017b)". Please explain what you mean, in order to have this manuscript self-standing. It's a bit more explicit in the conclusion but should be moved here.

P17L9: Additional sentence was included.

P18L2: "According to the findings in Sect. 3.2.2 that a large contribution of the unclosed energy balance is a missing sensible heat flux, we used the modeled data as a reference for the validation of the correction methods". I guess that the first part of the sentence should be deduced by the reader based on fig. 5. It would be better to comment this feature already in sect. 3.2.2. Also, I do not understand the link between the first part of the sentence and the second part. Whether is measured H or measured LE the main responsible for the non-closure of the energy balance, you can decide to use the modeled values as the reference. Please clarify.

P18L2: We deleted the link to Sect. 3.2.2 and hope it is now clear.

P18L15-15: "Additionally, this indicates that the Bowen-ratio correction is not a method that is applicable for the correction of the measurement errors that occur". Confusing since the reader does not know if this assertion holds for a given range of Bo or for the whole range.

P18L15: We have re-formulated.

P18L17-18: "Due to the assumption of a similarity between the water and carbon dioxide fluxes (Ruppert et al., 2006), neither the NEE flux nor the latent heat flux were corrected for high Bowen ratios". Not necessary to cite again Ruppert et al. at this stage. Also, this sentence is not well integrated in the discussion and is ambiguous. I would prefer something like: "In all cases, attribution of residual energy to latent heat flux is low for high bowen ratios. Therefore, due to the assumption of a similarity between the water and carbon dioxide fluxes, the NEE flux was only marginally corrected in these conditions".

P18L17-18: We have followed your suggestion.

P19L4-5: "Due to high Bowen ratios and large underestimation by the model, the buoyancy corrected fluxes show better results in comparison with the model". This sentence brings nothing in this discussion, I would simply delete it. If you think it brings necessary information, please improve it to be more explicit.

P19L4-5: was deleted.

P20L21-23: "The better correction...could be a reason". Not understandable for the reader. Please rephrase/improve. And consider revisiting the writing style of the whole conclusion to avoid this "telegraphic style".

P20L21-23: was deleted and a new sentence was added.

**Technical corrections:**

P1L16: replace "stratification" by "atmospheric stratification"

P1L16: was added.

P10L15: "of the" appears twice

P10L15: was done.

Fig 2: Distance units are missing

Fig. 2: was done.

Fig. 4: you can simplify the legend by "same as fig 3 but for the second GDP"

Fig 4: was done.

Reference:

Ruppert J, Thomas C and Foken T (2006) Scalar similarity for relaxed eddy accumulation methods. Boundary-Layer Meteorol. 120:39-63.

---

## Author Comment (AC4) · 28 Feb 2018

First of all we want to thank the three reviewers for the helpful comments. We have already answered the main problems of the reviewers in a letter to the editor on January 24, which was distributed to all reviewers: https://www.biogeosciences-discuss.net/bg-2017-450/bg-2017-450-AC1-supplement.pdf. Therefore, in the following we will answer only the specific problems. In accordance with these general comments we have made large additions tothe Introduction and deleted parts about the ACASA-model that were repetitions from earlier publications. Comments from referees are presented in black and these are followed by the authors' responses in blue.

**Reviewer #2**

According to your answer to the letter to the editor we have added a paragraph on advection in the introduction and have added some remarks in Sect. 3.2.4 about the relevance of our paper and the tile approach (this is also related to recommendations made by both of the other reviewers).

We believe that because of the high correlation coefficients in Tables 3 and 4, the result should not be questionable and a replacement by $R^2$ should not be necessary.

---

## Author Comment (AC5) · 28 Feb 2018

First of all we want to thank the three reviewers for the helpful comments. We have already answered the main problems of the reviewers in a letter to the editor on January 24, which was distributed to all reviewers: https://www.biogeosciences-discuss.net/bg-2017-450/bg-2017-450-AC1-supplement.pdf. Therefore, in the following we will answer only the specific problems. In accordance with these general comments we have made large additions tothe Introduction and deleted parts about the ACASA-model that were repetitions from earlier publications. Comments from referees are presented in black and these are followed by the authors' responses in blue.

**Reviewer #3**

**Special remarks:**

P1 L14 "models must be applied for different surface types" Comment: It is not quite clear why it is a MUST to apply any model at all and for different surfaces in particular?

P1L14:

Text was re-formulated and supplemented with additional text.

P1 L19 "not closed by an amount of up to 30%" Comment: 30% of unclosure is not an absolute maximum, is it?

P1L19: The paper is an overview paper.

P2 L2 "Measurement data must be corrected..." Comment: again – is it always a MUST?

P2L2: Text was reformulated.

P2 L25 "the footprint of different plant classes." Comment: What do you mean?

P2L25:

Text was re-formulated and supplemented with additional text.

P3 L9 "more detailed description of the surrounding topography is provided by Foken..." Comment: It is an important information for a study dealing with heterogeneous surface and should be provided in a manuscript.

P3L9: We believe that with Fig. 1a and 1b, the heights of the relevant hills in the surrounding, and the slope in the text, all necessary information is given.

P3 L12 "The main foliage is located" Comment: What is the main foliage? Do you mean a leaf area or mass or what?

P3L12: was re-formulated.

P3 L12 "where z is the measurement height normalized by the stand height hc" Comment: z is just a measurement height

P3L12: was corrected.

P4 Table 1. Why clearing is described in the table 1 and forest – in the text?

P4, Table 1: Table 1 includes the necessary data for the tile approach of the clearing. The modeling for the forest was done with the assumption of a homogeneous forest and the information is now also provided in Table 1.

P3 L12 "This method allows the calculation of the temperatures of these components without also making substantial errors in the case of significant deviations from the ambient temperature" Comment: 1) Why "also"? 2) Does it work for the components in a forest-clearing transition zone?

P5L12: The sentence was reformulated. It works for the clearing and the forest.

P5 L19 "Direct as well as diffuse radiation can be absorbed, transmitted, emitted or reflected by the canopy, whereby these processes are dependent on the leaf distribution of the plants." Comments: 1) The terms "direct" and "diffuse" are usually applied for shortwave solar (or diffuse sky) radiation fluxes which, therefore, cannot be emitted by the canopy under normal conditions. The canopy can emit a longwave radiation. 2) Why only the leaf distribution? What about stems, branches and soil?

P5L19: was corrected.

P5 L20 "the vegetation is distributed in 10 different leaf angle classes" Comments: the whole vegetation or just leaves/needles?

P5L20: This is now clarified in the text.

P5 L21 "allocation of the incoming energy" Comments: or of the available? The emitted/reflected parts of incoming energy are lost for the canopy.

P5L21: was corrected.

P5 L25 How tall is "taller vegetation"?

P5L25: was corrected.

P5 L26 What are the "diffusive approaches"?

P5L26: Reference was included and the sentence was reformulated, with a reference to the one-dimensional diffusion equation.

P6. Table 2: The measured parameters and instrumentation are not presented adequately: - What is the instrument for the air pressure? - CM14 is an albedometer and can measure both incoming and reflected shortwave radiation. Did you use the incoming component only? - CG2 – is the net radiometer, thus, the same question as for the CM14 - CNRN4 is probably the CNR4 net radiometer – the same questions about upward and downward fluxes. If you measured the downward fluxes only – how did you calculate the radiative balance? What is the instrument for the soil heat flux? Is there any instrument for PAR-measurements?

P6Table 2: The Table was corrected: pressure AB60; CM14, CG2: both components were used; CNR4: all four components were used; the "N" was deleted; soil heat flux was added; PAR see P7L18

P6 L4-5 "The soil respiration is separately calculated for 5 roots and microbes according to Eq. (1)" Comment: using the same R0 and Q10?

P7 L7 "the relation" – which relation?

P7 L13-14: As the information is important, I suggest to provide both parameter sets – clearing and forest for the better comparison and visualization of differences.

P6L4-5, P7L7¸ P7L13-14: This part was deleted because this was already published.

P7 L18 – PAR, measured or modelled? P7 L27 – Second reference of Haverd and Lindroth is not necessary P9 L3-4 "...but requires their scalar similarity. Henceforth, this method is abbreviated with EBC-Bo." Comment: The formulation is unclear – why scalar similarity produces this abbreviation? The same question is about EBC-HB.

P7L18: This part was deleted, because this was already published, PAR was not measured but calculated/parametrized from global radiation.

P7L27: was deleted.

P9L3-7: The Bowen ratio correction can only be applied when both fluxes (sensible and latent) are affected in a similar way by measuring errors or energy balance closure.

P9 L10 "Therefore, we propose a new correction method on the basis of the good scalar similarity between the humidity and the carbon dioxide concentration (Ruppert et al., 2006). Comment: Again, the formulation is unclear – is it your original suggestion or the one of the Ruppert et al?

P9L10: The correction for the latent heat flux can only be applied to the carbon dioxide flux if both fluxes are affected in a similar way by the energy balance closure problem. Ruppert et al. (2006) have shown that this required scalar similarity is not always valid, see Fig. 3 in this reference. An additional remark was included.

P9 L20 "of surface characteristics (roughness length)" – characteristic or characteristics?

P10 L22 "between the different stratification" – singular or plural?

P9L20, P10L22, was corrected.

P10 L24 " forest has additional influence" – how is it visible in Fig.2?

P10L24: The forest is within the footprint effect levels 80 and 95.

P11 Fig. 2: Consider placing the four (a) panels (upper row) below the four (b) panels (lower row) for a direct comparison of footprints. What is the "all" case?

P11Fig. 2: We have added some words for clarification.

P11 L7-8. The sentence is not clearly formulated please rewrite. What is the "land-use distribution of the whole clearing"? Do you mean different land covers or vegetation types?

P11L7-8: was corrected.

P11 L11-13"...difference...is more dominant" – please reformulate.

P11L11-13: was corrected.

P12L2-3 "Obviously, the integrated fluxes of the tile approaches for the whole clearing and the footprint of the turbulence mast for 5.5m height do not differ significantly" – what do you compare with what?

P12L2-3: Sentence has been corrected.

P12 L3: Which "land use characteristics"?

P12L3: was corrected.

P12 L4-5 "This can be seen by the high measured latent heat fluxes in the first Golden Day 5 Period (see Fig. 3). Further regression analyses..." Comment: Why "further" – there is no regression analysis on Fig.3?

P12L4-5: Further was deleted.

P12L3 Fig. 3. What about uncorrected fluxes?

P12L3, Fig 3: Uncorrected fluxes were not included because they are always smaller than modeled fluxes (Eq. 3).

P13 L3 "regression has been achieved" – can you achieve a regression?

P13 L3 "regression analysis is calculated for" – can you calculate an analysis?

P13L3: Both were corrected

P14 L1-2 "difference between net radiation and the ground heat flux" – how they were measured?

P14L1-2: The corrections in Table 2 should now make this clear, and a reference for ground heat flux has been included.

P14 L18 – "sensors" – which sensors?

P14L18: Has been added in Table 2.

P18 "Further research is necessary with other well parametrized models that close the energy balance very well, and data sets for lower Bowen ratios." Comment: does it mean that the present model is not adequate for the goal?

P18: We have made a small change in the sentence; we believe that the conclusions would be stronger if other models and data sets were to be analyzed in the same way.

P19 L2 "The comparison of the modeled and the measured (ground truth)" Comment: in previous section the ground truth was the model, wasn't it? P19 L5-6 "the buoyancy corrected fluxes show better results in comparison with the model. This offers the possibility of modeling fluxes over larger heterogeneous areas..." Comment: This does not look like a logical consequence for me.

P19L2: We have corrected this.

P20 L4 "can also be used with high accuracy for low vegetation if the plant specific parameters are appropriately implemented in the model" Comment: What in this case is the "high accuracy" and "low vegetation"? The conclusion does not follow from the results in my opinion.

P20L4: We have added some words for clarification.

P20 L7-12. The paragraph is not really clear. Try to reformulate it.

P20L7-12: We have added again the reference Falge et al (2017) for a better understanding.

Reference:

Ruppert J, Thomas C and Foken T (2006) Scalar similarity for relaxed eddy accumulation methods. Boundary-Layer Meteorol. 120:39-63.

---

## Author Response (AR2)

Dear Dr. Ibrom,
thank you very much for your comments as well as thanks also to the reviewers for validation of the revised manuscript and referee#1 for the comments to the revised manuscript . We have considered the general and specific points in the manuscript. In the following, we indicate the relevant changes in the manuscript according to the comments of the editor and the referee#1. Comments from the editor/referee are presented in black and these are followed by the authors' responses in blue.

**Editor:**

Dear Authors
I'm happy to tell that two of the referees indicated sufficient improvement of your latest revision and your explanations.

Please consider carefully the last remaining comments by Referee #1, mainly addressing some redundancy or still unclear parts of the text.

I'd like to especially encourage you to consider a change in the title that better reflects the novel parts of this study. The current title is rather *neutral* ~ *another application* and thus not as interesting as it could be.

We changed the title accordingly to: "Footprint weighted tile approach for a spruce forest and a nearby patchy clearing using the ACASA model".

Additionally in the abstract
line 2: add "homogeneous" before "spruce forests", to make the contrast of the objectives of the previous study to this study clearer. Then consider replacing 'the further application' with the novel scientific objective of the study, i.e. testing the model for heterogeneous vegetation.

"homogeneous" was included and application replaced with:  "testing the model using a footprint-weighted tile approach".

lines 6 and 7: To facilitate the understanding, consider replacing "Bowen ratio and a buoyancy flux correction method", which are not yet established terms, by writing what you actually did, e.g. "by forcing energy balance closure of the test data either by maintaining the measured Bowen ratio or by increasing the buoyancy flux, only"

The sentence was modified for better understanding:
"All measured fluxes are corrected by forcing energy balance closure of the test data either by maintaining the measured Bowen ratio or by the attribution of the residual depending on the fractions of sensible and latent heat flux to the buoyancy flux."

Please respond to every comment of the review and explain, how you have considered it in the revision of your manuscript and provide me with a track-changes and a final version of revision.

With kind regards,
Andreas Ibrom

**Reviewer #1**

The quality of the paper has been improved.

I'm still moderately enthusiastic about the originality level of the paper but at least specific advances compared to previous publications on the same site/model are more clearly presented. To this purpose, reorganization of the introduction section helps, even if I still have some remarks on it (see specific comments).

I still find the title very broad. It gives no idea of the precise scientific question you want to answer.

We changed the title accordingly to: "Footprint weighted tile approach for a spruce forest and a nearby patchy clearing using the ACASA model".

I agree with the authors' proposition of keeping the section 3.2.4. ("tile approach for a mixed forest site") now that the background is better explained, with the possibly biased use of remote sensing products (MODIS) vs flux dataset.

Even if most of my previous comments/suggestions have been taken into account, there are still some of them that where not really processed and when reading your new manuscript, they arose again in my mind so I simply reproduced them in the specific comments section after having adapted the line numbering for clarity sake.

See below for the answers to the specific comments.

Specific comments:

P1L13-P2L12: I'm again confused by your writing style. This is an important part of the paper and I think that the argumentation you gave in your first response to the editor (the part starting with "Reviewer #2 and #3 proposes the use of a 3D model rather than …" in your point4 was much clearer than what you have written here. I will therefore avoid detailed comments on this part and suggest revisiting it.

We have deleted a part of the introduction and have included some sentences of our answer to the reviewers #2 and #3:
"Experimental advection studies with an acceptable number of measurement points failed (Aubinet et al., 2010; Aubinet et al., 2012). Furthermore, the resolution of available 3-D models (Sogachev et al., 2002; Sogachev and Lloyd, 2004) is too large (25–50 m) for the small scale heterogeneities in the clearing (5–10 m). "

P2L24-29: "However, the influence of secondary structures on the energy balance cannot be taken into account by a one-dimensional model. The ACASA model is much better suited to handling coherent structures and counter gradients than classical SVAT models with a first order closure". This sounds like a repetition of what is written in the first part of the paragraph. Please merge the two. And the next sentence is not connected enough to/not explicit enough in the context of the § content.

We have deleted the first sentence that is not relevant in this part of the text.

P3L2: I would appreciate here a short comment on the dataset used, in order to help the reader understand why you need both datasets (and then understand better the current title).

We have moved a sentence from page 18 into the introduction to underline the need of both data sets:
"The inclusion of both surfaces in this study is necessary, because flux studies often use MODIS (Moderate-resolution Imaging Spectroradiometer) data with a grid size of 500 m or 250 m. Since the windthrow in 2007, the satellite sees about 50 % forest and 50 % clearing in the relevant grid."

P8L21: "The discrepancy between measured and simulated NEE can be an effect of the unclosed energy balance on the $CO_2$ fluxes". I understand what you mean but this is a complicated way of saying that if $CO_2$ exchanges share the same transport processes than heat exchanges (scalar similarity), measured $CO_2$ exchanges should be underestimated on the same level as heat fluxes. And this hypothesis being far from widely accepted, this point should be discussed further.

We have included one more reference and a sentence about the relevance of the method in relation to the present data handling:
"Therefore, we propose a new correction method on the basis of the good scalar similarity between the humidity and the carbon dioxide concentration mainly in the first part of the day, for scalar similarity see Ruppert et al. (2006), while the similarity between the temperature and other scalars often fails (Pearson et al., 1998). "

"The correction is not very large. Therefore, the up to now valid agreement of no application of an energy balance correction is reasonable."

P9L1: "whereby a spectral method of the non-linear flux averaging of surface characteristics (friction, roughness length) according to Hasager and Jensen (1999) is employed (Göckede et al., 2006)". I still do not understand this part of the sentence. Please be more explicit.

The Hasager & Jensen model is to complicate to explain it in one sentence, but we added a remark that linear averaging of the roughness length (input parameter of footprint models) is inaccurate:
"For the footprint calculations of this study, a Lagrangian footprint model is applied (Rannik et al., 2000, 2003), whereby a spectral method of the non-linear flux averaging of surface characteristics (friction, roughness length) according to Hasager and Jensen (1999) is employed (Göckede et al., 2006) because the linear averaging of the roughness length (input parameter of footprint models) is inaccurate. "

P9L14-18: This § should be reorganized to avoid repetitions.

We have slightly modified the sentence, but the effect of the three sectors N, NW, and NE must be explained:
"Regarding the stable stratified case, the north-westerly part of the spruce forest is

only included in the outer source region of the turbulent for flux 2.25m measuring height and is more pronounced for 5.5m measuring height. Additionally, a slight influence from the north-easterly spruce areas connected to the forest edge is calculated for 5.5m measuring height. "

P14L21-22: "For Bo > 1, the buoyancy correction overestimates the effect of thermal convection on the energy balance closure and the true correction might lie between both correction methods". I do not understand this sentence. Why EBC-HB should overestimate H when H is dominant? Please be more explicit.

We have modified the sentence:
"For Bo > 1, the buoyancy correction, which assumes that only convection is the reason of the unclosed energy balance, overestimates the energy balance closure correction and the true correction might lie between both correction methods. "

P17L8-9: "This could also be an overestimation by the measured fluxes due to the turbulence and the forest structure, discussed by Foken (2017b)". Please explain what you mean, in order to have this manuscript self-standing. It's a bit more explicit in the conclusion but should be moved here.

We have added some remarks that are published in both papers:
"This could be an overestimation by the measured fluxes due to the increased mechanical turbulence and consequently also turbulent fluxes caused by the heterogeneous forest structure, discussed by Foken (2017b). Furthermore, a related LES study, Kanani-Sühring and Raasch (2015) shows a maximal flux at a distance from the roughness step of about 10 times the canopy height that is nearly the location of the tower. "

P18L2: "Because a large contribution of the unclosed energy balance is a missing sensible heat flux, we used the modeled data, which close the energy balance by definition - as an etalon, for the validation of the correction methods". I do not understand the link between the first part of the sentence and the second part. Whether is measured H or measured LE the main responsible for the non-closure of the energy balance, you can decide to use the modeled values as the reference. Please clarify.

We have deleted one part of the sentence. This should clarify the text.
"In this section we used the modeled data, which close the energy balance by definition – as an etalon, for the validation of the correction methods. "

P18L14-15: "Therefore, the Bowen-ratio correction is not a method that is applicable for the correction of the measurement errors that occur". Confusing since the reader does not know if this assertion holds for a given range of Bo or for the whole range.

We have deleted the sentence.

Technical comments:

P23L6-7: here you use 2017a for the whole Foken book while on L23 of the same

page, you use it for a specific chapter. On the other hand, you use 2017b for another specific chapter on L26. Please standardize your choices.

We could not find that here is something not in agreement with the Copernicus styling file. Foken (2017a) is the whole book (Energy and Matter Fluxes of a Spruce Forest Ecosystem), while Foken (2017b), Foken et al. (2017a), and Foken et al. (2017b) are single chapters of the book. Foken (2017a) was not used in the text but only as a reference for the data set, which is documented in several chapters.

**New references:**

[revised manuscript text omitted]